# On Enforcing Better Conditioned Meta-Learning for Rapid Few-Shot Adaptation

**Markus Hiller[1]**    **Mehrtash Harandi[2]**    **Tom Drummond[1]**

[1]School of Computing and Information Systems, The University of Melbourne
[2]Department of Electrical and Computer Systems Engineering, Monash University
markus.hiller@student.unimelb.edu.au
mehrtash.harandi@monash.edu
tom.drummond@unimelb.edu.au

## Abstract

Inspired by the concept of preconditioning, we propose a novel method to increase adaptation speed for gradient-based meta-learning methods without incurring extra parameters. We demonstrate that recasting the optimisation problem to a non-linear least-squares formulation provides a principled way to actively enforce a *well-conditioned* parameter space for meta-learning models based on the concepts of the condition number and local curvature. Our comprehensive evaluations show that the proposed method significantly outperforms its unconstrained counterpart especially during initial adaptation steps, while achieving comparable or better overall results on several few-shot classification tasks – creating the possibility of dynamically choosing the number of adaptation steps at inference time.

## 1 Introduction

Learning a new task from only a very limited number of datapoints in the target domain poses a central challenge in machine learning. Gradient-based meta-learning as introduced in MAML (Finn et al., 2017) has proved to be a versatile tool to successfully address such *few-shot* problems, and has inspired a significant number of follow-up works tackling a diverse set of applications like regression (Nichol et al., 2018; Finn et al., 2018), image classification (Zintgraf et al., 2019; Antoniou et al., 2019; Rajeswaran et al., 2019), reinforcement (Al-Shedivat et al., 2018) and online continual learning (Gupta et al., 2020).

Broadly speaking, such approaches define a family of tasks and attempt to solve a bi-level optimization problem. The outer loop aims to learn an effective *meta initialization* of the parameters capturing properties that generalize well across different training tasks. This learned set of parameters is then *adapted* to a novel and previously unseen task in *only a few steps* during the inner loop.

Intuitively, one might expect that each step taken during the inner-loop adaptation ought to be as efficient as possible, enabling the model to quickly converge towards the desired goal (i.e., solving the new task). Evaluating popular methods like MAML (Finn et al., 2017) however shows that this is not the case (Figures 1 and 2). Particularly the initial steps fall short of this expectation. The reason for this behaviour is readily explained: Gradient directions do generally not point towards the actual desired minimum (Figure 1) as a result of the often ill-conditioned parameter space in which the optimization is performed.

This behaviour is undesirable for a number of reasons. Having high contribution towards reaching the optimum from the start opens up the possibility of dynamically choosing the number of steps to take at inference time, an interesting property for applications with limited or changing access to computational resources, and might further lead to a better overall convergence during the final steps.

36th Conference on Neural Information Processing Systems (NeurIPS 2022).

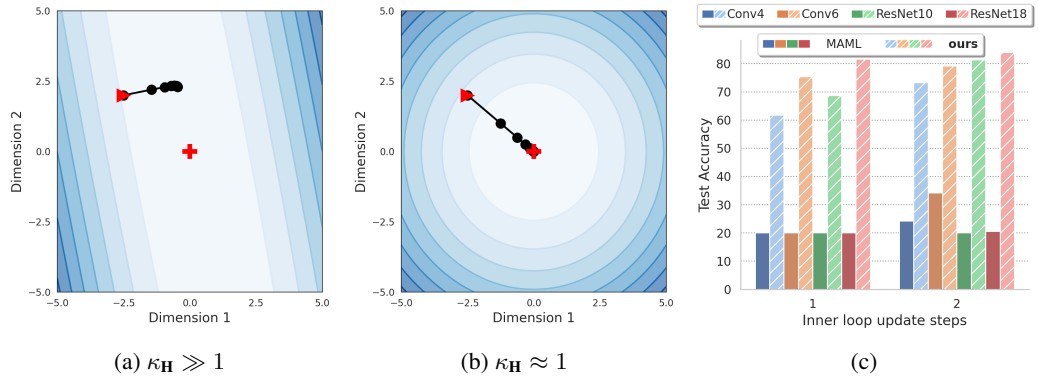

| | (a) $\kappa_{\mathbf{H}} \gg 1$ | (b) $\kappa_{\mathbf{H}} \approx 1$ | (c) |

Figure 1: **Importance of a well-conditioned parameter space.** (a) and (b) show the difference in convergence if optimization using SGD is performed in an ill-conditioned parameter space with high condition number $\kappa_{\mathbf{H}} \gg 1$ or well-conditioned one ($\kappa_{\mathbf{H}} \approx 1$), respectively. Depicted is an example for two dimensions using 10 update steps with learning rate 0.5. (c) Average accuracies on the CUB test set for the baseline *without* (MAML) and *with* our proposed conditioning constraint. Displayed are the first two out of five inner-loop update steps for different network architectures.

With this goal of rapid adaptation from the very first step in mind, we draw inspiration from the field of optimization – specifically from the idea of *preconditioning*. Instead of performing parameter updates directly, a suitable preconditioning matrix $\mathbf{P}$ can be chosen and multiplied to the gradient $\nabla_{\boldsymbol{\theta}}$, implicitly transforming the optimization problem to a space where updates can be performed more efficiently and resulting in a modulated parameter update

$$\boldsymbol{\theta}^{(k)} \;=\; \boldsymbol{\theta}^{(k-1)} - \alpha \, \mathbf{P} \, \nabla_{\boldsymbol{\theta}^{(k-1)}} \mathcal{L}\left(\mathcal{D}, f_{\boldsymbol{\theta}^{(k-1)}}\right) . \tag{1}$$

The reader might notice that this modified gradient descent update rule directly reduces to the well-known Newton's method if the preconditioning matrix is chosen to be the inverse of the Hessian matrix ($\mathbf{P} = \mathbf{H}^{-1}$ with $\mathbf{H} = \frac{\partial^2}{\partial \boldsymbol{\theta} \partial \boldsymbol{\theta}^{\top}}$) or an approximation thereof, and would thus enjoy a quadratic convergence rate via appropriate scaling of the gradients based on the local curvature. The actual choice and computation of a preconditioner is however non-trivial for most optimization problems, especially in the context of deep learning. Variants developed in recent years (Li et al., 2017; Gupta et al., 2018; Park & Oliva, 2019; Simon et al., 2020) can all be seen as proposing different ways of estimating suitable preconditioning matrices, often leading to a significant increase in parameters.

**Our method:** In contrast to previous works, we propose a new way to achieve preconditioning without incurring any additional parameters. Reconsidering the definition of preconditioned optimization stated in Equation (1), we raise the following hypothesis:

*If $f_{\boldsymbol{\theta}}$ is solely parameterized by $\boldsymbol{\theta}$ and $\boldsymbol{\theta}$ is learned, there exists a set of parameters $\tilde{\boldsymbol{\theta}}$ that achieves comparable performance on unseen tasks but possesses a low condition number $\kappa$ which allows fast convergence with gradient descent even when choosing the preconditioning matrix to $\tilde{\mathbf{P}} = \mathbf{I}$.*

In other words, the preconditioning matrix $\tilde{\mathbf{P}}$ can be absorbed into this 'new' learnable function $f_{\tilde{\boldsymbol{\theta}}}$, enabling the problem to be formulated and optimized directly in a well-conditioned space.

This raises the following questions: (i) Can the conditioning property of a model be actively influenced without negatively affecting its ability to converge and to solve a given task? (ii) If so, does this indeed lead to faster task-adaptation? In this paper, we set out to provide insights and answers to these and additional questions. In particular, our contributions include the following:

1. We show that the condition number of a network is a clear indicator for its few-step performance on an unseen task.
2. We demonstrate a principled way to derive the condition number of a network by reformulating the optimization problem to a non-linear least-squares form over the space of available samples in the support set.
3. We introduce a novel cost function based around the condition number to actively encourage learning of a well-conditioned parameter space for the model during training, allowing for rapid adaptation at inference time in very few steps.

4. We conduct in-depth analyses regarding different network architectures, ability to continue adaptation beyond the training horizon, performance regarding an extended set of $N$-way $K$-shot few-shot classification scenarios, and demonstrate the efficacy of our method on all five popular few-shot classification benchmarks.

## 2 Preliminaries

One highly influential algorithm that tackles few-shot learning in a gradient-based fashion has been *Model Agnostic Meta-Learning* (MAML) proposed by Finn et al. (2017). We start by introducing this algorithm to define the specific problem setting we are aiming to solve and to provide context for the following discussions. This is followed by a brief overview of the *condition number* which constitutes one of the main concepts our method is based on.

### 2.1 Problem setting and MAML

Let $\mathcal{D}_\tau^{\text{train}}$ and $\mathcal{D}_\tau^{\text{val}}$ be the training and validation set of some given task $\tau \sim p(\mathcal{T})$ (e.g. image classification), respectively. Further, assume that $\mathcal{D} := \{\boldsymbol{x}_i, y_i\}_{i=1}^{|\mathcal{D}|}$, with $\boldsymbol{x}_i \in \mathcal{X}, y_i \in \mathcal{Y}$ for some small $|\mathcal{D}|$. The predictor function of the model is denoted as $f : \mathcal{X} \times \mathbb{R}^n \to \mathcal{Y}$ and is parameterized by $\boldsymbol{\theta} \in \mathbb{R}^n$. MAML (Finn et al., 2017) then seeks to determine a universal meta initialization $\boldsymbol{\theta}^*$ by solving

$$\arg\min_{\boldsymbol{\theta}^*} \sum_{\tau \sim p(\mathcal{T})} \mathcal{L}\left(\mathcal{D}_\tau^{\text{val}}, \ \boldsymbol{\theta}_\tau^{(K)}\left(\mathcal{D}_\tau^{\text{train}}, \boldsymbol{\theta}^*\right)\right), \tag{2}$$

where the inner-loop task-specific parameters are given by

$$\boldsymbol{\theta}_\tau^{(k)} = \boldsymbol{\theta}_\tau^{(k-1)} - \alpha \nabla_{\boldsymbol{\theta}^{(k-1)}} \mathcal{L}\left(\mathcal{D}_\tau^{\text{train}}, \boldsymbol{\theta}_\tau^{(k-1)}\right) \tag{3}$$

for $k \in [1, 2, \ldots, K]$ with $\boldsymbol{\theta}_\tau^0 = \boldsymbol{\theta}^*$ and step size $\alpha$. In other words, given a task $\tau$ MAML starts from the meta-initialisation $\boldsymbol{\theta}^*$ and performs $K$ gradient update steps (so-called *inner-loop updates*) using the few available training samples of $\mathcal{D}_\tau^{\text{train}}$ for this specific task to obtain the adapted task-specific parameters $\boldsymbol{\theta}_\tau^{(K)}$ (Equation (3)). It then uses the unseen samples of $\mathcal{D}_\tau^{\text{val}}$ of this task together with the adapted parameters $\boldsymbol{\theta}_\tau^{(K)}$ to improve the meta-initialisation $\boldsymbol{\theta}^*$ in the so-called *outer-loop update* (Equation (2)). Note that this is possible since $\boldsymbol{\theta}_\tau^{(K)}$ is dependent on its initialisation $\boldsymbol{\theta}_\tau^0 = \boldsymbol{\theta}^*$ as defined in Equation (3), and gradients can thus be computed throughout all $K$ inner-loop updates.

### 2.2 Condition number of a matrix

One way to quantify local curvature of a parameter space and represent the expected convergence behaviour using a single scalar value is the *condition number* $\kappa_\mathbf{H}$ of the Hessian $\mathbf{H}_f$ of a function $f$ that is to be optimized. For a square-summable sequence space $\ell^2$ (i.e., Euclidean), the condition number of $\mathbf{H}_f$ can be computed via its maximum and minimum singular values $\sigma(\mathbf{H}_f)$, or its eigenvalues $\lambda(\mathbf{H}_f)$ in case $\mathbf{H}_f$ is normal, as

$$\kappa_\mathbf{H} = \frac{|\sigma_{\max}(\mathbf{H}_f)|}{|\sigma_{\min}(\mathbf{H}_f)|} \quad \text{or} \quad \kappa_\mathbf{H} = \frac{|\lambda_{\max}(\mathbf{H}_f)|}{|\lambda_{\min}(\mathbf{H}_f)|} \quad \text{if} \quad \mathbf{H}_f^* \mathbf{H}_f = \mathbf{H}_f \mathbf{H}_f^*. \tag{4}$$

The high condition number associated with the optimization problem depicted in Figure 1 (a) represents a higher difference in local curvature across the parameter space, leading to slow convergence caused by the initial gradients not pointing towards the minimum. In contrast, the better conditioned problem shown in (b) with a significantly lower condition number $\kappa_\mathbf{H} \approx 1$ approaches the desired minimum on a direct way in very few steps.

## 3 Learning a better-conditioned parameter space

We start this section by introducing how the objective used in few-shot learning can be reformulated as a linear least-squares problem, providing the basis to approximate its Hessian. We then introduce our proposed method to enforce a well conditioned parameter space and discuss a variation with constraining only a parameter subset that allows better scalability to models with higher parameter counts.

## 3.1 Reformulated problem setting

We reconsider the minimization objective of one inner-loop step $k$ used to perform the task-specific adaptation. The average loss over the support set samples in the inner loop can be reformulated to describe a non-linear least squares problem

$$\mathcal{L}\big(\mathcal{D}_\tau^{\text{train}}, \boldsymbol{\theta}_\tau^{(k)}\big) \;=\; \sum_{i=1}^{|\mathcal{D}_\tau^{\text{train}}|} \sqrt{\frac{1}{|\mathcal{D}_\tau^{\text{train}}|}\ell\left(\boldsymbol{x}_i, y_i, \boldsymbol{\theta}_\tau^{(k)}\right)}^2 \;=\; \sum_{i=1}^{|\mathcal{D}_\tau^{\text{train}}|} r_i\big(\boldsymbol{\theta}_\tau^{(k)}\big)^2, \tag{5}$$

with $r_i\big(\boldsymbol{\theta}_\tau^{(k)}\big)$ representing the residual of each individual sample $i$ and $|\mathcal{D}_\tau^{\text{train}}|$ the number of available task-specific samples.

As introduced in the previous section, the Hessian of a function describes its local curvature and can be used to gain insights into the convergence behaviour if the function is to be minimized via some gradient-based optimisation method (Figure 1). Note that the dependence of gradient, Jacobian and Hessian on the dataset $\mathcal{D}_\tau^{\text{train}}$ and parameters $\boldsymbol{\theta}_\tau^{(k)}$ is being partially dropped from the notation for improved readability during the following derivation. The dependence of the residual $r_i\big(\boldsymbol{\theta}_\tau^{(k)}\big)$ on the parameters $\boldsymbol{\theta}_\tau^{(k)}$ is additionally abbreviated via its simplified version $r_i(\boldsymbol{\theta})$, and $\mathcal{D}_\tau^{\text{train}}$ is referred to as $\mathcal{D}$. Using the residual-based notation of Equation (5) together with the introduced simplifications regarding notation, the gradient of $\mathcal{L}\big(\mathcal{D}_\tau^{\text{train}}, \boldsymbol{\theta}_\tau^{(k)}\big)$ with respect to some parameter $\theta_j$ is given by

$$g_j = 2\sum_{i=1}^{|\mathcal{D}|} r_i(\boldsymbol{\theta}) \frac{\partial r_i(\boldsymbol{\theta})}{\partial \theta_j}, \tag{6}$$

and the elements of its Hessian, computed by differentiating the gradient elements $g_j$ with respect to parameter $\theta_m$, are given by

$$H_{jm} = 2\sum_{i=1}^{|\mathcal{D}|} \left( \frac{\partial r_i(\boldsymbol{\theta})}{\partial \theta_j} \frac{\partial r_i(\boldsymbol{\theta})}{\partial \theta_m} + r_i(\boldsymbol{\theta}) \frac{\partial^2 r_i(\boldsymbol{\theta})}{\partial \theta_j \partial \theta_m} \right). \tag{7}$$

Ignoring the second-order derivative terms of the Hessian leads to the well-known *Gauss-Newton method* in which the Hessian is approximated via the use of the Jacobian in the form of

$$H_{jm} \approx 2\sum_{i=1}^{|\mathcal{D}|} \left( \frac{\partial r_i(\boldsymbol{\theta})}{\partial \theta_j} \frac{\partial r_i(\boldsymbol{\theta})}{\partial \theta_m} \right) = 2\sum_{i=1}^{|\mathcal{D}|} J_{ij} J_{im} \qquad \text{or} \qquad \mathbf{H} \approx 2\,\mathbf{J}^\top \mathbf{J}. \tag{8}$$

It is to be noted that we did not modify any component of the original problem formulation at any point. Thus, the Jacobian product approximating the Hessian of our objective function directly reflects the convergence behaviour of the optimisation problem. More specifically, computing the condition number $\kappa_{\mathbf{H}}$ (Section 2.2) provides a scalar measure how well- or ill-posed the problem is.

## 3.2 Conditioning constraint and overall objective

Computing the condition number by using the maximum and minimum eigenvalues as defined in Equation (4) ignores the distribution of the remaining $|\mathcal{D}_\tau^{\text{train}}|-2$ eigenvalues and would unnecessarily weaken the training signal. Instead we propose to use a measure considering the distribution of all eigenvalues $\boldsymbol{\lambda}(\mathbf{H})$ based on the observation that a well-conditioned problem with $\kappa_{\mathbf{H}} \approx 1$ can only be achieved if all eigenvalues are almost identical. We model this requirement by penalizing high variance of the approximated Hessian's logarithmic eigenvalues and define this *conditioning constraint* as a new loss function over the $K$ inner-loop updates as

$$\mathcal{L}_\kappa \left( \boldsymbol{\theta}_\tau^{(K)}\big(\mathcal{D}_\tau^{\text{train}}, \boldsymbol{\theta}^*\big) \right) = \frac{1}{K} \sum_{k=1}^{K} \text{Var}\left( \log_{10}\left( \boldsymbol{\lambda}\left( \mathbf{J}^{(k)} \mathbf{J}^{(k)\top} \right) \right) \right). \tag{9}$$

This formulation contains two design choices. The first choice to use logarithmic eigenvalues is founded in the fact that the actual magnitude is irrelevant for the convergence property since only its relative magnitude compared to all other eigenvalues is of importance. Using the logarithmic

Table 1: **Validating baseline re-implementation.** Results of our re-implementation vs. the original work of Finn et al. (2017). Means and 95% confidence intervals of 600 randomly generated test episodes for 5-way 1-shot and 5-way 5-shot classification on the *mini*ImageNet using a Conv4 model.

| | 5-shot | | 1-shot | |
|---|---|---|---|---|
| **Method** | **Reported** | **Ours** | **Reported** | **Ours** |
| MAML (Finn et al., 2017) | $63.11_{\pm 0.92}$ | $64.50_{\pm 0.69}$ | $48.70_{\pm 1.84}$ | $48.15_{\pm 0.80}$ |

form to compute the variance provides a scale-invariant way of penalising deviation and thus aligns with the ratio-based definition of the condition number, since it returns the same loss penalty for $\lambda_1 = 0.1, \lambda_2 = 0.2$ as for $\lambda_1 = 10, \lambda_2 = 20$ – both of which would have an identical condition number $\kappa = 2$ and thus be considered to have similar convergence properties. The second choice of equally weighting all $K$ inner-loop update steps is based on simplicity, and future research might explore whether non-equal weighting is beneficial.

Due to the number of samples being significantly smaller than the number of parameters and the resulting rank-deficiency of $\mathbf{J}^\top \mathbf{J}$, only a subset of size $|\mathcal{D}_\tau^{\text{train}}|$ of the total parameter space is directly influenced by the condition loss $\mathcal{L}_\kappa$ at each task iteration, whereas the remainder is its null space. This allows us to instead compute the smaller product $\mathbf{J}\mathbf{J}^\top$ with equivalent non-zero eigenvalues. It is to be noted that the gradients used for this computation will however backpropagate to all parameters involved in forming $\mathbf{J}$, thus appropriately constraining all relevant parameters.

The overall objective for the outer loop of the meta-learning process with weighting factor $\gamma$

$$\arg\min_{\boldsymbol{\theta}^*} \sum_{\tau \sim p(\mathcal{T})} \mathcal{L}\left(\mathcal{D}_\tau^{\text{val}}, \ \boldsymbol{\theta}_\tau^{(K)}\left(\mathcal{D}_\tau^{\text{train}}, \boldsymbol{\theta}^*\right)\right) \ + \ \gamma \mathcal{L}_\kappa\left(\boldsymbol{\theta}_\tau^{(K)}\left(\mathcal{D}_\tau^{\text{train}}, \boldsymbol{\theta}^*\right)\right) \tag{10}$$

encourages the network to learn a set of initialization parameters $\boldsymbol{\theta}^*$ that not only generalizes well across tasks ($\mathcal{L}$) but also possesses a well-posed problem structure with low condition number in the inner loop that allows rapid adaptation in very few steps ($\mathcal{L}_\kappa$). This is possible since the condition number representing the convergence quality can computed via the approximated Hessian which in turn is dependent on the network's parameters, allowing the computation of gradients all the way back to the initialization $\boldsymbol{\theta}^*$.

### 3.3 Constraining a parameter subset

To reduce complexity, improve scalability and considering the fact that the network might face some trade-off between accuracy and well-conditioned problem formulation induced by the objective in Equation (10), we build upon the findings of Raghu et al. (2019) that mainly the last few layers undergo significant adaptation during inner-loop updates. We thus propose and investigate constraining only a specific subset of the network's overall parameters through our proposed conditioning loss, namely the indicated 'last few layers'. This significantly reduces computational requirements and makes our approach more scalable to deeper networks. The only modification required is a slight adaptation of Equation (10) by replacing $\boldsymbol{\theta}_\tau^{(K)}$ in $\mathcal{L}_\kappa$ with the new parameter subset $\boldsymbol{\psi}_\tau^{(K)} \subseteq \boldsymbol{\theta}_\tau^{(K)}$.

## 4 Experimental evaluation

In our experimental evaluation, we aim to gain empirical insights to answer the following questions: (i) Can the conditioning property of a model be actively influenced without negatively affecting its ability to converge and to solve a task? (ii) If so, does this indeed lead to faster task-adaptation? (iii) How does it impact networks of various depths and trained on different datasets? (iv) Does a better-conditioned parameter space provide advantages across different many-way multi-shot settings? (v) Can we expect continued adaptation beyond the training horizon?

### 4.1 Experimental setup

We choose MAML (Finn et al., 2017) as our baseline for comparison due to its vast popularity and the influence regarding follow-up works it inspired since its introduction (see Section 6). To validate

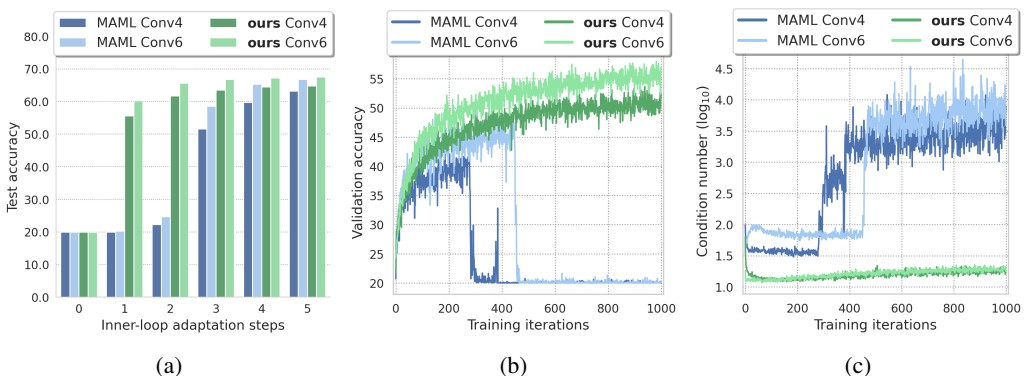

Figure 2: **Enforcing well-conditioned adaptation.** Comparison of baseline method MAML (Finn et al., 2017) and the same method with our proposed conditioning constraint. Results were obtained on *tiered*Imagenet for two different architectures. (a) Test accuracy over inner-loop adaptation steps. (b) Validation accuracy of inner-loop adaptation step 1 during training. (c) Condition number of parameters at adaptation step 1 during training, depicted in logarithmic scale.

the implementation of our MAML baseline, we report the results achieved by the original and our re-implemented baseline in Table 1, showing results are on-par with the original version.

We evaluate our method on all five popular few-shot classification datasets, namely *mini*ImageNet (Vinyals et al., 2016; Ravi & Larochelle, 2017), *tiered*ImageNet (Ren et al., 2018), CIFAR-FS (Bertinetto et al., 2019), FC100 (Oreshkin et al., 2018) and CUB-200-2011 (Wah et al., 2011), hereafter referred to as 'CUB'. We follow previous work like Chen et al. (2019) and report the averaged test accuracies of 600 randomly sampled experiments using the $k$ samples of the respective $k$-shot setting as support and 16 query samples for each class. Our baselines and methods are trained on a single NVIDIA RTX 3090 for 60,000 episodes for 1-shot and 40,000 for 5-shot tasks, with the best model picked based on highest validation accuracy. We use 5 inner-loop updates and $\gamma = 1$.

### 4.2 Condition number indicates few-step performance

As very first point, we provide insights into the basis this work is built on: The condition number of a network is closely related to its adaptation capabilities using very few steps. To this end, we take a closer look at the behaviour of a network's condition number during training and the associated performance in terms of validation accuracy after the first inner-loop update step (out of five), for methods trained *without* or *with* our proposed conditioning objective (Equation (10)). The results depicted in Figure 2 demonstrate a clear dependence between the progress of the validation accuracy (b) and its associated condition number (c) in two ways: 1) Low condition number seems to generally align with high validation accuracy, and 2) sudden increases in condition number indicate instability regarding the validation accuracy. We further observe that the lower condition numbers of both models trained with the proposed constraint ('*ours*') seem to improve convergence speed early on during training, leading to a slightly steeper increase in validation accuracy.

This dependence between condition number and validation accuracy exists across all inner-loop update steps, with corresponding discussion provided in the supplementary material of this paper.

### 4.3 Per-step adaptation with enforced conditioning

Having demonstrated the existing dependence between the condition number and performance of a model, we now concentrate on our hypothesis raised in the introduction that actively constraining the network's parameter space to enforce better conditioning is possible without negatively affecting overall performance on unseen tasks. In Figure 2 (c) can be clearly observed that models trained *with* $\mathcal{L}_\kappa$ ('*ours*') exhibit significantly lower condition numbers than their unconstrained counterparts ('*MAML*'), demonstrating that our proposed loss is indeed able to actively encourage learning of a better-conditioned parameter space.

The performance of these models achieved on novel tasks of the test set is depicted across all five inner-loop update steps in Figure 2 (a) for a 5-way 5-shot scenario. While all networks seem to learn

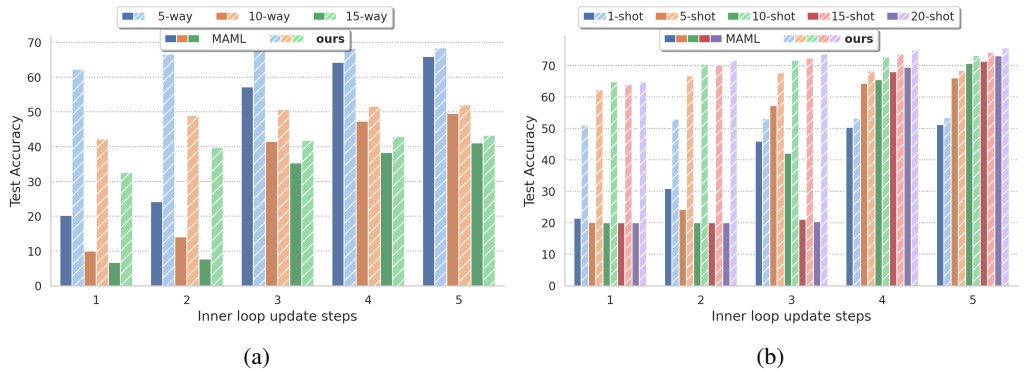

Figure 3: **Many-way multi-shot scenarios.** Reported are the average accuracy for the baseline *without* conditioning (MAML) and the version *with* our proposed conditioning loss on the *mini*ImageNet testset. (a) depicts the results for various $N$-way 5-shot, (b) for 5-way $K$-shot scenarios.

a similarly general meta initialization (step 0), the effect of using our proposed $\mathcal{L}_\kappa$ is significant especially during the first steps, increasing the test accuracy by $35.68\%$ and $39.48\%$ (Conv4), and $39.98\%$ and $40.95\%$ (Conv6) for adaptation steps 1 and 2, respectively. While the unconstrained models mostly catch up during the following steps, their adaptation behaviour prevents any use of such trained models with fewer than the maximum number of adaptation steps used in training (here 5) – in stark contrast to our models which achieve $86\%$ (Conv4) and $89\%$ (Conv6) of their top accuracy already after one single step. We further observed that the faster initial convergence of our method seems to consistently lead to slightly higher overall performance across most experiments we performed, especially for deeper architectures (Section 4.5) with $K \geq 5$.

## 4.4 Effect of constrained parameter subsets

We now take a closer look at two distinct questions: (i) is the condition number of a parameter subset 'representative' of the actual condition number of the network (based on all parameters) and if so, (ii) how do different subsets influence the networks adaptation performance? To this end, we train a Conv4 architecture and enforce conditioning on various parameter subsets of the last few layers. The development of the average condition numbers during training with respect to only the parameters of the classifier ('*cls*') and to all parameters of the network ('*all*') indicate that the condition number of this subset is indeed representative for the entire network (Figure 4), an observation that applies to all evaluated subsets except the parameters of the embedding layer's batchnorm ('*eBN*'). For further discussion, please see the supplementary material. Table 2 shows that all methods demonstrate significant improvements of test accuracy in the first few steps compared to the unconstrained baseline, and all except '*eBN*' achieve superior overall performance after 5 steps. Despite these results indicating the slight advantage of selecting subset '*emb*', we conduct the following experiments using the '*cls*' subset for the reason that constraining the parameters of the classifier is independent of the backbone and allows for easier transition between different models.

## 4.5 Increased network depths and adaptation beyond the training horizon

We conduct an extensive set of experiments with four different architectures (Conv4, Conv6, ResNet10, ResNet18) comparing the baseline and our method across all 5 datasets to evaluate the performance for changing network depths, both in 1-shot and 5-shot settings. The results in Table 4 demonstrate the efficacy of the proposed constraint across all models (please refer to the supplementary material for complete results). We additionally observe that the better-conditioned parameter space seems to also encourage convergence to a better overall optimum especially for deeper architectures, e.g. ResNet18 with $73.28\%$ vs. baseline $71.06\%$. Interestingly, smaller unconstrained baseline models like Conv4 seem to naturally learn a better conditioned parameter space in 1-shot scenarios for most datasets, which might indicate some relationship between the condition number and the combination of task difficulty, gradient noise and network capacity. Additional results in the lower part of Table 4 show that while our methods are trained using five inner-loop update steps and achieve very fast adaptation from the very first step onwards, they are additionally able to continuously adapt further beyond their training horizon when given the possibility.

Table 2: **Influence of condition parameter selection** on test accuracy (CUB dataset, Conv4). The condition loss $\mathcal{L}_\kappa$ is enforced w.r.t. one or multiple parameter groups. *cls*: classifier (last layer): *emb*: embedding layer; *eBN*: BN of embedding layer. Highlighted in gray is the 'default' setting used throughout this work (simplicity).

| $\mathcal{L}_\kappa$ w.r.t. | | | 5-way 5-shot | | | | |
|---|---|---|---|---|---|---|---|
| cls | emb | eBN | step 1↑ | step 2↑ | step 3↑ | step 4↑ | step 5↑ |
| | | | $20.04_{\pm0.02}$ | $24.16_{\pm0.38}$ | $64.30_{\pm0.84}$ | $74.71_{\pm0.78}$ | $77.06_{\pm0.69}$ |
| ✓ | | | $61.78_{\pm0.75}$ | $73.28_{\pm0.71}$ | $75.64_{\pm0.71}$ | $76.75_{\pm0.68}$ | $77.24_{\pm0.67}$ |
| | ✓ | | $59.85_{\pm0.80}$ | $\mathbf{73.54_{\pm0.75}}$ | $\mathbf{76.61_{\pm0.69}}$ | $\mathbf{77.73_{\pm0.67}}$ | $\mathbf{78.14_{\pm0.67}}$ |
| | | ✓ | $62.22_{\pm0.79}$ | $69.87_{\pm0.79}$ | $73.04_{\pm0.77}$ | $74.15_{\pm0.74}$ | $74.86_{\pm0.72}$ |
| ✓ | ✓ | | $\mathbf{64.27_{\pm0.77}}$ | $73.45_{\pm0.71}$ | $76.02_{\pm0.69}$ | $77.30_{\pm0.67}$ | $77.77_{\pm0.68}$ |
| | ✓ | ✓ | $55.54_{\pm0.83}$ | $71.63_{\pm0.76}$ | $75.55_{\pm0.71}$ | $76.78_{\pm0.70}$ | $77.12_{\pm0.68}$ |
| ✓ | | ✓ | $63.46_{\pm0.76}$ | $73.49_{\pm0.74}$ | $76.31_{\pm0.69}$ | $77.67_{\pm0.67}$ | $78.10_{\pm0.67}$ |
| ✓ | ✓ | ✓ | $60.93_{\pm0.81}$ | $71.39_{\pm0.75}$ | $74.83_{\pm0.71}$ | $76.40_{\pm0.69}$ | $77.28_{\pm0.70}$ |

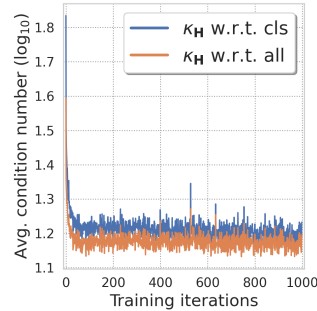

Figure 4: Average condition number during training w.r.t *cls* subset and *all* parameters.

## 4.6 Many-way and multi-shot scenarios

We finally investigate the influence of a better conditioned parameter space for $N$-way $K$-shot applications with more than the 'default' $N=5$ classes and $K>5$ support samples. Figure 3 (a) demonstrates that our method sustains the previously demonstrated fast adaptation from the first step on and seems to handle settings with the increased difficulty of more classes slightly better than the baseline. Scenarios that provide more than 5 samples per class (Figure 3 (b)) show that while our method continues to consistently outperform the baseline by significant margins during the first few steps (e.g. by $53\%$ for 20-shot at step 3), the convergence of the unconstrained baseline seems to get delayed even further with increasing sample numbers – indicating that the conditioning property can not be improved simply through more available data, but is in contrast rather adversely affected.

## 4.7 Preconditioning methods – Number of parameters and performance

We compare our approach to other recently-published preconditioning methods for a 5-way 5-shot classification task evaluated on the *mini*ImageNet test set. In this context, we want to draw attention the fact that while most published methods compare based on the used backbone, it is worth considering their introduction of additional parameters that are required at both training and inference time. We therefore outline the gains in accuracy in relation to the total number of parameters in Figure 5. While all methods are able to improve the test accuracy upon their respective baselines, this comes at a cost. Meta-SGD (Li et al., 2017) is able to improve by $1.5\%$ relative to its Conv4 (32) MAML baseline (Finn et al., 2017) but doubles the number of parameters (+100%). ModGrad (Simon et al., 2020) is able to notably

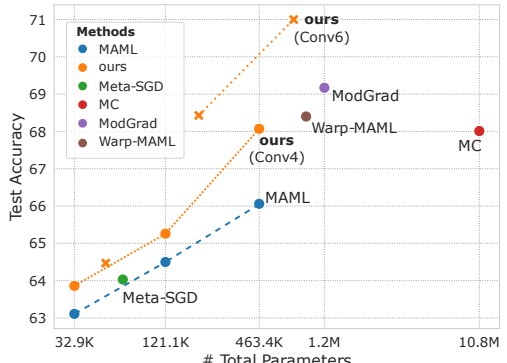

Figure 5: **Preconditioning methods, number of parameters and accuracies** obtained for 5-way 5-shot evaluated on the *mini*ImageNet test set.

outperform its Conv4 (64) baseline by $7.2\%$ but requires an increase in parameters of +873%, while Warp-MAML (Flennerhag et al., 2019) and MC (Park & Oliva, 2019) achieve their relative improvements of $3.5\%$ and $3.0\%$ at the cost of incurring additional parameter costs of +96% and +2235%, respectively (both Conv4 (128)). Our approach in contrast improves upon all baselines without adding any additional parameters – a property that allows us to instead choose larger backbones like Conv6 and thus significantly outperform other methods across the entire parameter-vs.-accuracy spectrum. Please refer to the supplementary material for more detailed information.

## 4.8 Taking fewer but bigger steps at inference time

Our motivational example depicted in Figure 1 implies that well-conditioned parameter space should provide a more 'direct' path towards the optimum and thus enable us to take fewer but bigger steps.

We therefore investigate the behaviour of our method when taking *only one step* but with a *greater step size* at inference time. Note that the network has been trained as in previous experiments with the default setting of using 5 inner-loop adaptation steps with a learning rate 'lr' during training.

Our obtained results (Table 3) clearly indicate that a better conditioned parameter space does indeed provide the ability to take fewer but bigger steps towards the minimum to adapt faster at inference time. While our approach evaluated on the same setting as used during training (5 steps with 'lr') still slightly outperforms its 1-step variants and thus hints at a not (yet) ideally-conditioned parameter space, our method taking only 1 step with an appropriately scaled '5·lr' step size demonstrates promising task adaptation and even outperforms the respective 5-step MAML baseline model.

Table 3: **Varying number of steps and step size at inference time.** Test accuracy for 5-way 5-shot on *mini*ImageNet. Model has been trained with default settings of 5 inner-loop update steps with learning rate lr. Reported are the classification accuracies on the test set, averaged over 600 tasks.

| Conv6 | 0 steps | 1 step - lr | 1 step - 2·lr | 1 step - 3·lr | 1 step - 4·lr | 1 step - 5·lr | 5 steps - lr |
|---|---|---|---|---|---|---|---|
| MAML | $20.00_{\pm 0.00}$ | $20.19_{\pm 0.07}$ | $20.00_{\pm 0.00}$ | $20.14_{\pm 0.05}$ | $23.28_{\pm 0.37}$ | $26.64_{\pm 0.51}$ | $65.96 \pm 0.71$ |
| ours | $20.14_{\pm 0.60}$ | $62.31_{\pm 0.72}$ | $65.67_{\pm 0.71}$ | $65.87_{\pm 0.71}$ | $65.72_{\pm 0.71}$ | $66.54_{\pm 0.70}$ | $68.43 \pm 0.71$ |

## 5   Limitations

Applying our proposed condition loss in its naive form to all parameters of very deep networks might be infeasible due to high parameter counts involved in computing the Jacobians. Our experiments however clearly demonstrate that constraining a representative subset like the classifier is sufficient to improve the overall conditioning property of the model and can significantly improve the convergence behaviour during initial steps (Section 4.4).

Our method further makes use of second order derivatives (just like the baseline MAML) and thus needs to store the computational graph throughout the inner-loop updates, a property that didn't raise any issues in our experiments but might be a point of concern for applications with extremely limited memory – a general point worth considering with second-order methods.

## 6   Related work

**Gradient-based meta-learning**.  Methods most related to our approach generally aim to find a set of model parameters that can adapt to novel tasks within a few steps of gradient descent. Early seminal works (Andrychowicz et al., 2016; Finn et al., 2017) have spiked significant follow-up research like first order variants (Nichol et al., 2018), probabilistic reformulations (Finn et al., 2018) and generative approaches (Rusu et al., 2018), among many more (Ravi & Larochelle, 2017; Rajeswaran et al., 2019; Zintgraf et al., 2019). While some have derived theoretical guarantees (Balcan et al., 2019; Grant et al., 2018), others improved training stability (Antoniou et al., 2019). Two main challenges have to be overcome to solve a previously unseen task with a highly limited number of data samples in this way: overfitting (Sun et al., 2019) and rapid adaptation. While several works have attempted the first (Rusu et al., 2018; Zhou et al., 2018; Zintgraf et al., 2019), learning an efficient way of updating the model's parameters that enables fast adaptation remains an open problem (Flennerhag et al., 2019). On common drawback is the unknown adaptation quality, i.e. how many updates have to be performed to achieve satisfactory results on a new task. This is commonly solved by either simply choosing the de-facto default value of five steps, or the highest number of steps that seems feasible for a given task based on empirical results. In contrast, our proposed approach of constraining methods to learn a well-conditioned parameter space achieves fast adaptation from the very first step onwards and gradually increases, allowing users to dynamically choose the number at inference time.

**Gradient modulation**. Few works have attempted to tackle the challenge of finding a more suitable way of updating the parameters. Early methods use LSTMs to update models iteratively by learning both the initialisation and update strategy (Andrychowicz et al., 2016; Ravi & Larochelle, 2017) but are difficult to train (Li et al., 2017). This idea has been further developed by Li et al. (2017) towards Meta-SGD, where the initialisation is meta-learned together with update direction and parameter-specific learning rates. While this can be seen as a form of preconditioned optimisation, more direct attempts have recently been proposed. An interleaving-based method is used to improve

Table 4: **Increasing the network depth and adaptation steps.** Evaluations are conducted on all five popular FSL datasets: CUB-200-2011 (Wah et al., 2011), *mini*ImageNet (Vinyals et al., 2016), *tiered*ImageNet (Ren et al., 2018), CIFAR-FS (Bertinetto et al., 2019) and FC100 (Oreshkin et al., 2018). Reported are the classification accuracies on the unseen test set, averaged over 600 tasks.

| | Network | Method | 5-shot | | | | | 1-shot | | | | |
|---|---|---|---|---|---|---|---|---|---|---|---|---|
| | | | step 1↑ | step 2↑ | step 3↑ | step 4↑ | step 5↑ | step 1↑ | step 2↑ | step 3↑ | step 4↑ | step 5↑ |
| *mini*ImageNet | Conv4 | MAML | $20.00_{\pm0.01}$ | $20.00_{\pm0.00}$ | $52.87_{\pm0.75}$ | $61.12_{\pm0.75}$ | $64.50_{\pm0.69}$ | $38.76_{\pm0.73}$ | $42.33_{\pm0.74}$ | $45.99_{\pm0.77}$ | $47.67_{\pm0.81}$ | $48.15_{\pm0.80}$ |
| | | ours | $56.09_{\pm0.66}$ | $62.28_{\pm0.70}$ | $64.01_{\pm0.67}$ | $64.89_{\pm0.67}$ | $65.26_{\pm0.67}$ | $43.68_{\pm0.74}$ | $47.65_{\pm0.79}$ | $48.50_{\pm0.79}$ | $48.76_{\pm0.79}$ | $48.94_{\pm0.80}$ |
| | Conv6 | MAML | $20.19_{\pm0.07}$ | $24.17_{\pm0.39}$ | $57.25_{\pm0.72}$ | $64.24_{\pm0.73}$ | $65.96_{\pm0.71}$ | $21.40_{\pm0.23}$ | $30.85_{\pm0.66}$ | $45.91_{\pm0.87}$ | $50.30_{\pm0.88}$ | $51.22_{\pm0.88}$ |
| | | ours | $62.31_{\pm0.72}$ | $66.66_{\pm0.71}$ | $67.63_{\pm0.72}$ | $68.21_{\pm0.71}$ | $68.43_{\pm0.71}$ | $50.92_{\pm0.85}$ | $52.98_{\pm0.89}$ | $53.18_{\pm0.89}$ | $53.28_{\pm0.89}$ | $53.34_{\pm0.89}$ |
| | ResNet10 | MAML | $20.06_{\pm0.05}$ | $20.03_{\pm0.04}$ | $41.74_{\pm0.74}$ | $63.28_{\pm0.72}$ | $69.43_{\pm0.71}$ | $20.02_{\pm0.04}$ | $22.14_{\pm0.30}$ | $50.99_{\pm0.84}$ | $56.47_{\pm0.82}$ | $57.35_{\pm0.80}$ |
| | | ours | $53.93_{\pm0.75}$ | $68.96_{\pm0.72}$ | $71.40_{\pm0.70}$ | $72.18_{\pm0.69}$ | $72.46_{\pm0.67}$ | $50.44_{\pm0.82}$ | $55.86_{\pm0.84}$ | $57.44_{\pm0.84}$ | $57.97_{\pm0.85}$ | $58.20_{\pm0.85}$ |
| | ResNet18 | MAML | $20.12_{\pm0.06}$ | $20.00_{\pm0.00}$ | $36.64_{\pm0.69}$ | $65.01_{\pm0.74}$ | $71.06_{\pm0.74}$ | $20.02_{\pm0.03}$ | $20.79_{\pm0.17}$ | $52.81_{\pm0.86}$ | $56.42_{\pm0.90}$ | $56.84_{\pm0.90}$ |
| | | ours | $68.60_{\pm0.71}$ | $72.05_{\pm0.70}$ | $72.93_{\pm0.69}$ | $73.10_{\pm0.69}$ | $73.28_{\pm0.68}$ | $54.46_{\pm0.90}$ | $57.01_{\pm0.91}$ | $57.31_{\pm0.91}$ | $57.52_{\pm0.91}$ | $57.64_{\pm0.91}$ |

| | Dataset | Method | step 1↑ | step 2↑ | step 3↑ | step 4↑ | step 5↑ | step 1↑ | step 2↑ | step 3↑ | step 4↑ | step 5↑ |
|---|---|---|---|---|---|---|---|---|---|---|---|---|
| ResNet18 | CUB | MAML | $20.00_{\pm0.00}$ | $20.48_{\pm0.13}$ | $43.96_{\pm0.86}$ | $79.56_{\pm0.74}$ | $83.56_{\pm0.61}$ | $20.02_{\pm0.02}$ | $25.57_{\pm0.54}$ | $72.20_{\pm0.98}$ | $74.06_{\pm0.95}$ | $74.52_{\pm0.94}$ |
| | | ours | $81.57_{\pm0.61}$ | $84.00_{\pm0.55}$ | $84.47_{\pm0.54}$ | $84.63_{\pm0.54}$ | $84.86_{\pm0.54}$ | $66.55_{\pm1.03}$ | $72.93_{\pm0.98}$ | $74.38_{\pm0.97}$ | $74.96_{\pm0.97}$ | $75.22_{\pm0.97}$ |
| | *ti*ImgNet | MAML | $20.00_{\pm0.01}$ | $20.10_{\pm0.05}$ | $38.93_{\pm0.79}$ | $68.57_{\pm0.88}$ | $73.90_{\pm0.79}$ | $20.02_{\pm0.02}$ | $21.19_{\pm0.23}$ | $51.30_{\pm0.95}$ | $56.80_{\pm1.00}$ | $57.71_{\pm1.00}$ |
| | | ours | $70.13_{\pm0.78}$ | $73.60_{\pm0.74}$ | $74.31_{\pm0.74}$ | $74.55_{\pm0.74}$ | $74.67_{\pm0.74}$ | $54.79_{\pm0.96}$ | $57.44_{\pm0.97}$ | $57.64_{\pm0.97}$ | $57.71_{\pm0.98}$ | $57.80_{\pm0.98}$ |
| | CIFAR-FS | MAML | $20.01_{\pm0.01}$ | $20.04_{\pm0.03}$ | $45.75_{\pm0.83}$ | $75.69_{\pm0.81}$ | $79.59_{\pm0.70}$ | $20.00_{\pm0.00}$ | $23.41_{\pm0.39}$ | $63.34_{\pm0.99}$ | $67.05_{\pm0.97}$ | $67.44_{\pm0.99}$ |
| | | ours | $77.31_{\pm0.71}$ | $79.49_{\pm0.67}$ | $79.96_{\pm0.66}$ | $80.09_{\pm0.65}$ | $80.18_{\pm0.65}$ | $65.73_{\pm1.03}$ | $68.39_{\pm1.04}$ | $68.75_{\pm1.04}$ | $69.04_{\pm1.03}$ | $69.09_{\pm1.03}$ |
| | FC100 | MAML | $20.00_{\pm0.00}$ | $20.00_{\pm0.00}$ | $20.00_{\pm0.00}$ | $48.46_{\pm0.73}$ | $48.56_{\pm0.76}$ | $20.00_{\pm0.00}$ | $20.04_{\pm0.02}$ | $32.30_{\pm0.64}$ | $34.37_{\pm0.69}$ | $35.19_{\pm0.70}$ |
| | | ours | $47.11_{\pm0.73}$ | $50.16_{\pm0.73}$ | $50.82_{\pm0.74}$ | $51.08_{\pm0.74}$ | $51.22_{\pm0.74}$ | $33.24_{\pm0.69}$ | $36.05_{\pm0.71}$ | $36.70_{\pm0.72}$ | $36.90_{\pm0.72}$ | $37.02_{\pm0.72}$ |

| | Network | Method | step 5↑ | step 10↑ | step 25↑ | step 50↑ | step 100↑ | step 5↑ | step 10↑ | step 25↑ | step 50↑ | step 100↑ |
|---|---|---|---|---|---|---|---|---|---|---|---|---|
| CUB | Conv4 | ours | $77.24_{\pm0.67}$ | $77.48_{\pm0.68}$ | $77.74_{\pm0.67}$ | $77.88_{\pm0.68}$ | $78.06_{\pm0.68}$ | $63.26_{\pm0.98}$ | $63.42_{\pm0.99}$ | $63.44_{\pm1.00}$ | $63.54_{\pm1.00}$ | $63.47_{\pm1.00}$ |
| | Conv6 | ours | $80.65_{\pm0.63}$ | $80.90_{\pm0.63}$ | $81.15_{\pm0.62}$ | $81.25_{\pm0.62}$ | $81.35_{\pm0.62}$ | $68.87_{\pm1.06}$ | $69.12_{\pm1.06}$ | $69.19_{\pm1.06}$ | $69.16_{\pm1.06}$ | $69.15_{\pm1.06}$ |
| | ResNet10 | ours | $83.82_{\pm0.59}$ | $84.09_{\pm0.58}$ | $84.11_{\pm0.59}$ | $84.26_{\pm0.57}$ | $84.28_{\pm0.57}$ | $74.99_{\pm0.96}$ | $74.99_{\pm0.97}$ | $75.19_{\pm0.97}$ | $75.24_{\pm0.97}$ | $75.33_{\pm0.97}$ |
| | ResNet18 | ours | $84.66_{\pm0.54}$ | $84.83_{\pm0.53}$ | $84.94_{\pm0.53}$ | $85.01_{\pm0.53}$ | $85.01_{\pm0.53}$ | $75.22_{\pm0.97}$ | $75.52_{\pm0.95}$ | $75.66_{\pm0.95}$ | $75.74_{\pm0.95}$ | $75.81_{\pm0.95}$ |

the convergence speed via meta-learning a parameterized preconditioning matrix by Flennerhag et al. (2019), while Park & Oliva (2019) instead learn curvature information to transform the gradients driving the task-specific adaptation and Simon et al. (2020) modulate the gradients using an additional module and low-rank approximation. However, all these approaches aim to estimate some form of additional preconditioning matrix to transform the gradients, incurring an extensive number of additional parameters and overhead at both training and inference time. In contrast, our method imposes a constraint directly onto the parameter space of our network and thus targets the same idea of a well-conditioned optimisation space but within the already existing capacity of the network.

**Condition number.** The relevance of the condition number has been discussed in the context of deep learning by Goodfellow et al. (2016), and few works have made use of it as a 'passive metric', e.g. to monitor and assess quality of convergence in the context of Generative Adversarial Networks (Zuo et al., 2021). We take a different approach and show that in the bi-level nature of gradient-based few-shot meta-learning, we can actively manipulate the learning process of the function to learn a well-posed optimisation problem that demonstrates superior convergence properties.

# 7 Conclusion

In this paper, we demonstrated how to actively encourage a deep network to learn a better-conditioned parameter space that allows significantly faster adaptation to novel tasks in few-shot settings. To this end, we introduced a new constraint that is easily integrated into optimisation-based few-shot methods via an additive loss term. Our constraint is based around the condition number of local curvatures and does not incur any additional parameters. We showed the efficacy of our approach through comprehensive experiments on five datasets with various different architectures and few-show scenarios, significantly outperforming unconstrained methods during initial adaptation.

**Acknowledgements.** The authors would like to thank Finn et al. (2017) and Chen et al. (2019) for sharing their insights, experimental details and code.

Part of this research was conducted while MH was with Monash University. This research was partially undertaken using the LIEF HPC-GPGPU Facility hosted at the University of Melbourne. This Facility was established with the assistance of LIEF Grant LE170100200.

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
