# On Enforcing Better Conditioned Meta-Learning for Rapid Few-Shot Adaptation

## Supplementary Material

**Markus Hiller[1]    Mehrtash Harandi[2]    Tom Drummond[1]**

[1]School of Computing and Information Systems, The University of Melbourne
[2]Department of Electrical and Computer Systems Engineering, Monash University
markus.hiller@student.unimelb.edu.au
mehrtash.harandi@monash.edu
tom.drummond@unimelb.edu.au

## A    Datasets used for experiments

This section provides additional background information about the five datasets we use to train and evaluate the baseline and our approach in the main paper.

**CUB-200-2011**. The Caltech-UCSD Birds-200-2011, *aka* CUB-200-2011 or simply 'CUB', is focused on fine-grained image classification tasks. It was proposed by Wah et al. (2011) and contains 11,788 images of 200 subcategories. We follow previous works like Chen et al. (2019) and use the evaluation protocol introduced by Hilliard et al. (2018), splitting the dataset into 100 classes for training, 50 for validation and 50 for testing.

*mini***ImageNet**. The *mini*ImageNet dataset has been initially proposed by (Vinyals et al., 2016) and the specific few-shot settings have been further refined in later work by (Ravi & Larochelle, 2017). It consists of a 100 class subset selected from the ImageNet dataset (Russakovsky et al., 2015) with 600 images for each class. The dataset is split into 64 training, 16 validation and 20 test classes.

*tiered***ImageNet**. The *tiered*ImageNet (Ren et al., 2018) dataset is equally a subset of classes selected from the bigger ImageNet (Russakovsky et al., 2015) dataset, however with a different structure and substantially larger set of classes. It is composed of 34 super-classes with a total of 608 categories that are split into 20, 6 and 8 super-classes totalling in 779,165 images. This unique split aims at achieving better separation between training, validation and testing, respectively.

**CIFAR-FS**. The CIFAR-FS dataset (Bertinetto et al., 2019) contains essentially the data from the CIFAR100 (Krizhevsky et al., 2009) dataset and splits the 100 categories of 600 images each into 64 training, 16 validation and 20 test classes.

**FC-100**. The FC-100 dataset (Oreshkin et al., 2018) is similarly derived from CIFAR100 (Krizhevsky et al., 2009) and split into 60 training, 20 validation and 20 test classes, but follows a splitting approach more similar to *tiered*ImageNet to increase separation between classes and difficulty.

## B    Effect of increased network depth

All experiments are conducted with equal contribution of all adaptation steps to the conditioning loss (as defined in Equation (9) of the main paper), with the conditioning constraint enforced with respect to the parameters of the classifier. To provide insights into the effect of increasing the depth and number of parameters onto the conditioning performance, we evaluate the following architectures on all five popular few-shot classification benchmarks for 5-shot and 1-shot settings: Convolutional networks with 4 layers (Conv4) and 6 layers (Conv6), as well as the two residual networks ResNet10

36th Conference on Neural Information Processing Systems (NeurIPS 2022).

and ResNet18. While a selection of the results has been discussed in the main paper, the test accuracies on all datasets across all architectures are presented in Table A1.

Table A1: **Increasing the network's depth and number of parameters.** Evaluations are conducted on all five popular FSL datasets: CUB-200-2011 (Wah et al., 2011), *mini*ImageNet (Vinyals et al., 2016), *tiered*ImageNet (Ren et al., 2018), CIFAR-FS (Bertinetto et al., 2019) and FC100 (Oreshkin et al., 2018). Reported are the classification accuracies on the unseen test set, averaged over 600 tasks following previous works like Chen et al. (2019).

| | Network | Method | 5-shot | | | | | 1-shot | | | | |
|---|---|---|---|---|---|---|---|---|---|---|---|---|
| | | | step 1↑ | step 2↑ | step 3↑ | step 4↑ | step 5↑ | step 1↑ | step 2↑ | step 3↑ | step 4↑ | step 5↑ |
| CUB | Conv4 | MAML | $20.04_{\pm0.02}$ | $24.16_{\pm0.38}$ | $64.30_{\pm0.84}$ | $74.71_{\pm0.78}$ | $77.06_{\pm0.69}$ | $50.53_{\pm0.92}$ | $57.94_{\pm0.97}$ | $60.40_{\pm0.98}$ | $60.90_{\pm0.98}$ | $61.09_{\pm0.98}$ |
| | | ours | $61.78_{\pm0.75}$ | $73.28_{\pm0.71}$ | $75.64_{\pm0.71}$ | $76.75_{\pm0.68}$ | $77.24_{\pm0.67}$ | $55.93_{\pm0.92}$ | $61.62_{\pm0.96}$ | $62.82_{\pm0.98}$ | $63.20_{\pm0.99}$ | $63.26_{\pm0.98}$ |
| | Conv6 | MAML | $20.00_{\pm0.00}$ | $34.14_{\pm0.62}$ | $73.58_{\pm0.82}$ | $79.40_{\pm0.70}$ | $80.52_{\pm0.65}$ | $20.06_{\pm0.04}$ | $25.85_{\pm0.48}$ | $62.54_{\pm1.02}$ | $66.78_{\pm0.98}$ | $67.67_{\pm0.98}$ |
| | | ours | $75.42_{\pm0.72}$ | $79.21_{\pm0.65}$ | $80.01_{\pm0.65}$ | $80.44_{\pm0.63}$ | $80.65_{\pm0.63}$ | $65.15_{\pm1.06}$ | $68.28_{\pm1.06}$ | $68.54_{\pm1.07}$ | $68.76_{\pm1.06}$ | $68.87_{\pm1.06}$ |
| | ResNet10 | MAML | $20.00_{\pm0.00}$ | $20.00_{\pm0.00}$ | $20.60_{\pm0.13}$ | $76.65_{\pm0.79}$ | $82.13_{\pm0.64}$ | $20.00_{\pm0.00}$ | $33.55_{\pm0.79}$ | $67.41_{\pm1.08}$ | $72.20_{\pm0.97}$ | $73.04_{\pm0.96}$ |
| | | ours | $68.71_{\pm0.79}$ | $81.36_{\pm0.64}$ | $82.98_{\pm0.61}$ | $83.53_{\pm0.60}$ | $83.82_{\pm0.59}$ | $63.53_{\pm1.05}$ | $71.56_{\pm1.02}$ | $73.87_{\pm1.08}$ | $74.72_{\pm0.97}$ | $74.99_{\pm0.96}$ |
| | ResNet18 | MAML | $20.00_{\pm0.00}$ | $20.48_{\pm0.13}$ | $43.96_{\pm0.86}$ | $79.56_{\pm0.74}$ | $83.56_{\pm0.61}$ | $20.02_{\pm0.02}$ | $25.57_{\pm0.54}$ | $72.20_{\pm0.98}$ | $74.06_{\pm0.95}$ | $74.52_{\pm0.94}$ |
| | | ours | $81.57_{\pm0.61}$ | $84.00_{\pm0.55}$ | $84.47_{\pm0.54}$ | $84.63_{\pm0.54}$ | $84.66_{\pm0.54}$ | $66.55_{\pm1.03}$ | $72.93_{\pm0.98}$ | $74.38_{\pm0.97}$ | $74.96_{\pm0.97}$ | $75.22_{\pm0.97}$ |
| *mini*ImageNet | Conv4 | MAML | $20.00_{\pm0.01}$ | $20.00_{\pm0.00}$ | $52.87_{\pm0.75}$ | $61.12_{\pm0.75}$ | $64.50_{\pm0.69}$ | $38.76_{\pm0.73}$ | $42.33_{\pm0.74}$ | $45.99_{\pm0.77}$ | $47.67_{\pm0.81}$ | $48.15_{\pm0.80}$ |
| | | ours | $56.09_{\pm0.66}$ | $62.28_{\pm0.70}$ | $64.01_{\pm0.67}$ | $64.89_{\pm0.67}$ | $65.26_{\pm0.67}$ | $43.68_{\pm0.74}$ | $47.65_{\pm0.79}$ | $48.50_{\pm0.79}$ | $48.76_{\pm0.79}$ | $48.94_{\pm0.80}$ |
| | Conv6 | MAML | $20.19_{\pm0.07}$ | $24.17_{\pm0.39}$ | $57.25_{\pm0.72}$ | $64.24_{\pm0.73}$ | $65.96_{\pm0.71}$ | $21.40_{\pm0.23}$ | $30.85_{\pm0.66}$ | $45.91_{\pm0.87}$ | $50.30_{\pm0.88}$ | $51.22_{\pm0.88}$ |
| | | ours | $62.31_{\pm0.72}$ | $66.66_{\pm0.71}$ | $67.63_{\pm0.72}$ | $68.21_{\pm0.71}$ | $68.43_{\pm0.71}$ | $50.92_{\pm0.85}$ | $52.98_{\pm0.89}$ | $53.18_{\pm0.89}$ | $53.28_{\pm0.89}$ | $53.34_{\pm0.89}$ |
| | ResNet10 | MAML | $20.06_{\pm0.05}$ | $20.03_{\pm0.04}$ | $41.74_{\pm0.74}$ | $63.28_{\pm0.72}$ | $69.43_{\pm0.71}$ | $20.02_{\pm0.04}$ | $22.14_{\pm0.30}$ | $50.99_{\pm0.84}$ | $56.47_{\pm0.82}$ | $57.35_{\pm0.80}$ |
| | | ours | $53.93_{\pm0.75}$ | $68.96_{\pm0.72}$ | $71.40_{\pm0.70}$ | $72.18_{\pm0.69}$ | $72.46_{\pm0.67}$ | $50.44_{\pm0.82}$ | $55.86_{\pm0.84}$ | $57.44_{\pm0.84}$ | $57.97_{\pm0.85}$ | $58.20_{\pm0.85}$ |
| | ResNet18 | MAML | $20.12_{\pm0.06}$ | $20.00_{\pm0.00}$ | $36.64_{\pm0.69}$ | $65.01_{\pm0.74}$ | $71.06_{\pm0.74}$ | $20.02_{\pm0.03}$ | $20.79_{\pm0.17}$ | $52.81_{\pm0.86}$ | $56.42_{\pm0.90}$ | $56.84_{\pm0.90}$ |
| | | ours | $68.60_{\pm0.71}$ | $72.05_{\pm0.70}$ | $72.93_{\pm0.69}$ | $73.10_{\pm0.69}$ | $73.28_{\pm0.68}$ | $54.46_{\pm0.90}$ | $57.01_{\pm0.91}$ | $57.31_{\pm0.91}$ | $57.52_{\pm0.91}$ | $57.64_{\pm0.91}$ |
| *tiered*ImageNet | Conv4 | MAML | $20.00_{\pm0.00}$ | $22.30_{\pm0.28}$ | $51.64_{\pm0.77}$ | $59.76_{\pm0.83}$ | $63.26_{\pm0.77}$ | $40.25_{\pm0.80}$ | $45.17_{\pm0.85}$ | $46.49_{\pm0.89}$ | $47.03_{\pm0.89}$ | $47.33_{\pm0.90}$ |
| | | ours | $55.68_{\pm0.75}$ | $61.78_{\pm0.77}$ | $63.49_{\pm0.77}$ | $64.49_{\pm0.75}$ | $64.77_{\pm0.75}$ | $42.23_{\pm0.83}$ | $45.97_{\pm0.88}$ | $46.90_{\pm0.88}$ | $47.22_{\pm0.88}$ | $47.34_{\pm0.88}$ |
| | Conv6 | MAML | $20.29_{\pm0.10}$ | $24.73_{\pm0.42}$ | $58.56_{\pm0.79}$ | $65.33_{\pm0.82}$ | $66.78_{\pm0.79}$ | $43.29_{\pm0.91}$ | $48.13_{\pm0.92}$ | $49.59_{\pm0.95}$ | $50.20_{\pm0.94}$ | $50.40_{\pm0.96}$ |
| | | ours | $60.27_{\pm0.78}$ | $65.68_{\pm0.79}$ | $66.76_{\pm0.79}$ | $67.24_{\pm0.79}$ | $67.60_{\pm0.79}$ | $46.10_{\pm0.88}$ | $49.59_{\pm0.93}$ | $50.44_{\pm0.95}$ | $50.72_{\pm0.95}$ | $50.87_{\pm0.95}$ |
| | ResNet10 | MAML | $20.01_{\pm0.01}$ | $20.27_{\pm0.10}$ | $36.18_{\pm0.76}$ | $67.61_{\pm0.90}$ | $73.03_{\pm0.85}$ | $20.00_{\pm0.00}$ | $24.68_{\pm0.50}$ | $52.25_{\pm0.92}$ | $57.34_{\pm0.94}$ | $57.80_{\pm0.94}$ |
| | | ours | $58.15_{\pm0.83}$ | $71.83_{\pm0.75}$ | $74.40_{\pm0.74}$ | $75.40_{\pm0.72}$ | $75.77_{\pm0.71}$ | $52.88_{\pm0.97}$ | $58.09_{\pm0.96}$ | $59.19_{\pm0.96}$ | $59.54_{\pm0.96}$ | $59.65_{\pm0.96}$ |
| | ResNet18 | MAML | $20.00_{\pm0.01}$ | $20.10_{\pm0.05}$ | $38.93_{\pm0.79}$ | $68.57_{\pm0.88}$ | $73.90_{\pm0.79}$ | $20.02_{\pm0.02}$ | $21.19_{\pm0.23}$ | $51.30_{\pm0.95}$ | $56.80_{\pm1.00}$ | $57.71_{\pm1.00}$ |
| | | ours | $70.13_{\pm0.78}$ | $73.60_{\pm0.74}$ | $74.31_{\pm0.74}$ | $74.55_{\pm0.74}$ | $74.67_{\pm0.74}$ | $54.79_{\pm0.96}$ | $57.44_{\pm0.97}$ | $57.64_{\pm0.97}$ | $57.71_{\pm0.98}$ | $57.80_{\pm0.98}$ |
| CIFAR-FS | Conv4 | MAML | $20.00_{\pm0.01}$ | $20.00_{\pm0.00}$ | $56.12_{\pm0.85}$ | $67.08_{\pm0.81}$ | $69.97_{\pm0.75}$ | $34.71_{\pm0.75}$ | $42.34_{\pm0.91}$ | $48.57_{\pm0.96}$ | $51.10_{\pm0.95}$ | $51.84_{\pm0.93}$ |
| | | ours | $60.77_{\pm0.82}$ | $67.55_{\pm0.79}$ | $69.04_{\pm0.77}$ | $69.90_{\pm0.77}$ | $70.32_{\pm0.76}$ | $46.59_{\pm0.92}$ | $50.84_{\pm0.98}$ | $51.59_{\pm0.98}$ | $51.92_{\pm0.98}$ | $52.01_{\pm0.99}$ |
| | Conv6 | MAML | $20.22_{\pm0.08}$ | $24.66_{\pm0.40}$ | $65.34_{\pm0.85}$ | $72.33_{\pm0.82}$ | $74.00_{\pm0.79}$ | $24.83_{\pm0.47}$ | $43.05_{\pm0.82}$ | $54.90_{\pm0.95}$ | $57.11_{\pm0.95}$ | $57.83_{\pm0.95}$ |
| | | ours | $67.12_{\pm0.79}$ | $72.63_{\pm0.79}$ | $73.75_{\pm0.77}$ | $74.24_{\pm0.77}$ | $74.47_{\pm0.78}$ | $54.07_{\pm0.92}$ | $57.11_{\pm0.94}$ | $57.68_{\pm0.96}$ | $57.92_{\pm0.96}$ | $58.12_{\pm0.96}$ |
| | ResNet10 | MAML | $20.04_{\pm0.03}$ | $20.00_{\pm0.00}$ | $40.23_{\pm0.87}$ | $71.29_{\pm0.87}$ | $77.12_{\pm0.77}$ | $20.00_{\pm0.00}$ | $57.47_{\pm0.97}$ | $63.57_{\pm1.01}$ | $64.93_{\pm1.00}$ | $65.42_{\pm0.98}$ |
| | | ours | $60.93_{\pm0.87}$ | $75.65_{\pm0.76}$ | $77.67_{\pm0.72}$ | $78.45_{\pm0.72}$ | $78.90_{\pm0.70}$ | $57.58_{\pm0.96}$ | $63.97_{\pm0.94}$ | $65.63_{\pm0.94}$ | $66.07_{\pm0.94}$ | $66.17_{\pm0.95}$ |
| | ResNet18 | MAML | $20.01_{\pm0.01}$ | $20.04_{\pm0.03}$ | $45.75_{\pm0.83}$ | $75.69_{\pm0.81}$ | $79.59_{\pm0.70}$ | $20.00_{\pm0.00}$ | $20.04_{\pm0.39}$ | $63.34_{\pm0.99}$ | $67.05_{\pm0.97}$ | $67.44_{\pm0.99}$ |
| | | ours | $77.31_{\pm0.71}$ | $79.49_{\pm0.67}$ | $79.96_{\pm0.66}$ | $80.09_{\pm0.65}$ | $80.18_{\pm0.65}$ | $65.73_{\pm1.03}$ | $68.39_{\pm1.04}$ | $68.75_{\pm1.04}$ | $69.04_{\pm1.03}$ | $69.09_{\pm1.03}$ |
| FC100 | Conv4 | MAML | $20.00_{\pm0.00}$ | $20.00_{\pm0.00}$ | $38.03_{\pm0.66}$ | $42.96_{\pm0.74}$ | $47.61_{\pm0.72}$ | $20.04_{\pm0.03}$ | $27.98_{\pm0.57}$ | $31.12_{\pm0.68}$ | $35.17_{\pm0.76}$ | $36.08_{\pm0.76}$ |
| | | ours | $40.46_{\pm0.68}$ | $44.43_{\pm0.72}$ | $46.90_{\pm0.72}$ | $47.51_{\pm0.73}$ | $48.01_{\pm0.73}$ | $33.84_{\pm0.72}$ | $36.16_{\pm0.75}$ | $36.68_{\pm0.76}$ | $36.77_{\pm0.76}$ | $36.82_{\pm0.76}$ |
| | Conv6 | MAML | $20.00_{\pm0.00}$ | $20.70_{\pm0.14}$ | $39.62_{\pm0.59}$ | $42.72_{\pm0.71}$ | $46.48_{\pm0.70}$ | $30.31_{\pm0.66}$ | $32.49_{\pm0.69}$ | $34.67_{\pm0.74}$ | $35.23_{\pm0.76}$ | $35.64_{\pm0.76}$ |
| | | ours | $41.16_{\pm0.62}$ | $45.25_{\pm0.68}$ | $46.57_{\pm0.70}$ | $47.13_{\pm0.71}$ | $47.45_{\pm0.72}$ | $33.55_{\pm0.64}$ | $35.55_{\pm0.65}$ | $36.15_{\pm0.67}$ | $36.36_{\pm0.66}$ | $36.40_{\pm0.67}$ |
| | ResNet10 | MAML | $20.00_{\pm0.00}$ | $20.00_{\pm0.01}$ | $30.54_{\pm0.58}$ | $42.68_{\pm0.71}$ | $47.03_{\pm0.73}$ | $20.00_{\pm0.00}$ | $20.74_{\pm0.16}$ | $33.19_{\pm0.66}$ | $34.99_{\pm0.73}$ | $36.33_{\pm0.73}$ |
| | | ours | $42.66_{\pm0.68}$ | $48.58_{\pm0.70}$ | $49.94_{\pm0.69}$ | $50.65_{\pm0.69}$ | $50.86_{\pm0.70}$ | $32.17_{\pm0.63}$ | $35.63_{\pm0.71}$ | $36.78_{\pm0.72}$ | $37.31_{\pm0.73}$ | $37.50_{\pm0.73}$ |
| | ResNet18 | MAML | $20.00_{\pm0.00}$ | $20.00_{\pm0.00}$ | $20.00_{\pm0.00}$ | $48.46_{\pm0.73}$ | $48.56_{\pm0.76}$ | $20.00_{\pm0.00}$ | $20.04_{\pm0.02}$ | $32.30_{\pm0.64}$ | $34.37_{\pm0.69}$ | $35.19_{\pm0.70}$ |
| | | ours | $47.11_{\pm0.73}$ | $50.16_{\pm0.73}$ | $50.82_{\pm0.74}$ | $51.08_{\pm0.74}$ | $51.22_{\pm0.74}$ | $33.24_{\pm0.69}$ | $36.05_{\pm0.71}$ | $36.70_{\pm0.72}$ | $36.90_{\pm0.72}$ | $37.02_{\pm0.72}$ |

# C Adaptation beyond the training horizon

In this section, we provide further details regarding the behaviour of the baseline trained *without* ('MAML') and *with* the proposed condition loss $\mathcal{L}_\kappa$ ('ours') when the models are provided with the possibility to perform an increased number of adaptation steps beyond the training horizon **at test time only** – in our evaluated case up to 100 update steps. The training was in contrast performed with five adaptation steps. Results obtained on the test datasets for 5-way 5-shot and 5-way 1-shot scenarios are presented in Table A2 and follow the trends that have been discussed in the main paper.

Table A2: **Adaptation beyond the training horizon.** Evaluations are conducted on all five popular FSL datasets: CUB-200-2011 (Wah et al., 2011), *mini*ImageNet (Vinyals et al., 2016), *tiered*ImageNet (Ren et al., 2018), CIFAR-FS (Bertinetto et al., 2019) and FC100 (Oreshkin et al., 2018). Models have been trained with 5 inner-loop update steps, but are evaluated using additional update steps at inference time.

| | Network | Method | 5-shot | | | | | 1-shot | | | | |
| --- | --- | --- | --- | --- | --- | --- | --- | --- | --- | --- | --- | --- |
| | | | step 5↑ | step 10↑ | step 25↑ | step 50↑ | step 100↑ | step 5↑ | step 10↑ | step 25↑ | step 50↑ | step 100↑ |
| **CUB** | Conv4 | MAML | $77.06_{\pm0.69}$ | $77.84_{\pm0.66}$ | $77.87_{\pm0.65}$ | $77.78_{\pm0.66}$ | $77.83_{\pm0.65}$ | $61.09_{\pm0.98}$ | $61.33_{\pm0.99}$ | $61.50_{\pm0.99}$ | $61.61_{\pm0.98}$ | $61.65_{\pm0.98}$ |
| | | ours | $77.24_{\pm0.67}$ | $77.48_{\pm0.68}$ | $77.74_{\pm0.67}$ | $77.88_{\pm0.68}$ | $78.06_{\pm0.68}$ | $63.26_{\pm0.98}$ | $63.42_{\pm0.99}$ | $63.44_{\pm1.00}$ | $63.54_{\pm1.00}$ | $63.47_{\pm1.00}$ |
| | Conv6 | MAML | $80.52_{\pm0.65}$ | $80.89_{\pm0.63}$ | $81.00_{\pm0.63}$ | $81.00_{\pm0.63}$ | $81.01_{\pm0.64}$ | $67.67_{\pm0.98}$ | $68.03_{\pm0.98}$ | $68.39_{\pm0.98}$ | $68.48_{\pm0.98}$ | $68.56_{\pm0.98}$ |
| | | ours | $80.65_{\pm0.63}$ | $80.90_{\pm0.63}$ | $81.15_{\pm0.62}$ | $81.25_{\pm0.62}$ | $81.35_{\pm0.62}$ | $68.87_{\pm1.06}$ | $69.12_{\pm1.06}$ | $69.19_{\pm1.06}$ | $69.16_{\pm1.06}$ | $69.15_{\pm1.06}$ |
| | ResNet10 | MAML | $82.13_{\pm0.64}$ | $83.90_{\pm0.59}$ | $83.99_{\pm0.59}$ | $84.09_{\pm0.59}$ | $84.08_{\pm0.59}$ | $73.04_{\pm0.96}$ | $73.56_{\pm0.96}$ | $73.80_{\pm0.96}$ | $73.97_{\pm0.96}$ | $74.01_{\pm0.96}$ |
| | | ours | $83.82_{\pm0.59}$ | $84.09_{\pm0.58}$ | $84.11_{\pm0.59}$ | $84.26_{\pm0.57}$ | $84.28_{\pm0.57}$ | $74.99_{\pm0.96}$ | $74.99_{\pm0.97}$ | $75.19_{\pm0.97}$ | $75.24_{\pm0.97}$ | $75.33_{\pm0.97}$ |
| | ResNet18 | MAML | $83.56_{\pm0.61}$ | $84.52_{\pm0.55}$ | $84.56_{\pm0.55}$ | $84.65_{\pm0.54}$ | $84.59_{\pm0.56}$ | $74.52_{\pm0.94}$ | $74.97_{\pm0.95}$ | $75.23_{\pm0.95}$ | $75.30_{\pm0.95}$ | $75.34_{\pm0.94}$ |
| | | ours | $84.66_{\pm0.54}$ | $84.83_{\pm0.53}$ | $84.94_{\pm0.53}$ | $85.01_{\pm0.53}$ | $85.01_{\pm0.53}$ | $75.22_{\pm0.97}$ | $75.52_{\pm0.95}$ | $75.66_{\pm0.95}$ | $75.74_{\pm0.95}$ | $75.81_{\pm0.95}$ |
| **mini-ImageNet** | Conv4 | MAML | $64.50_{\pm0.69}$ | $65.35_{\pm0.70}$ | $65.71_{\pm0.70}$ | $65.90_{\pm0.70}$ | $66.07_{\pm0.70}$ | $48.15_{\pm0.80}$ | $48.52_{\pm0.80}$ | $48.75_{\pm0.79}$ | $48.86_{\pm0.79}$ | $48.90_{\pm0.79}$ |
| | | ours | $65.26_{\pm0.67}$ | $65.84_{\pm0.67}$ | $66.26_{\pm0.67}$ | $66.46_{\pm0.67}$ | $66.58_{\pm0.66}$ | $48.94_{\pm0.80}$ | $49.16_{\pm0.80}$ | $49.31_{\pm0.81}$ | $49.45_{\pm0.80}$ | $49.53_{\pm0.80}$ |
| | Conv6 | MAML | $65.96_{\pm0.71}$ | $66.55_{\pm0.71}$ | $66.63_{\pm0.72}$ | $66.72_{\pm0.71}$ | $66.80_{\pm0.71}$ | $51.22_{\pm0.88}$ | $51.42_{\pm0.87}$ | $51.48_{\pm0.87}$ | $51.52_{\pm0.87}$ | $51.53_{\pm0.87}$ |
| | | ours | $68.43_{\pm0.71}$ | $68.84_{\pm0.71}$ | $69.11_{\pm0.70}$ | $69.28_{\pm0.70}$ | $69.39_{\pm0.70}$ | $53.34_{\pm0.89}$ | $53.42_{\pm0.89}$ | $53.51_{\pm0.89}$ | $53.55_{\pm0.89}$ | $53.55_{\pm0.89}$ |
| | ResNet10 | MAML | $69.43_{\pm0.71}$ | $71.90_{\pm0.68}$ | $72.29_{\pm0.66}$ | $72.28_{\pm0.67}$ | $72.21_{\pm0.67}$ | $57.35_{\pm0.80}$ | $57.80_{\pm0.83}$ | $57.86_{\pm0.84}$ | $57.91_{\pm0.84}$ | $58.17_{\pm0.85}$ |
| | | ours | $72.46_{\pm0.71}$ | $73.10_{\pm0.67}$ | $73.33_{\pm0.68}$ | $73.28_{\pm0.68}$ | $73.35_{\pm0.68}$ | $58.20_{\pm0.85}$ | $58.28_{\pm0.87}$ | $58.27_{\pm0.87}$ | $58.29_{\pm0.87}$ | $58.38_{\pm0.88}$ |
| | ResNet18 | MAML | $71.06_{\pm0.74}$ | $73.37_{\pm0.68}$ | $73.57_{\pm0.69}$ | $73.56_{\pm0.69}$ | $73.53_{\pm0.69}$ | $56.84_{\pm0.90}$ | $57.11_{\pm0.91}$ | $56.97_{\pm0.91}$ | $56.96_{\pm0.91}$ | $57.00_{\pm0.91}$ |
| | | ours | $73.28_{\pm0.68}$ | $73.46_{\pm0.68}$ | $73.57_{\pm0.68}$ | $73.57_{\pm0.68}$ | $73.62_{\pm0.68}$ | $57.64_{\pm0.91}$ | $57.82_{\pm0.90}$ | $57.92_{\pm0.91}$ | $58.02_{\pm0.91}$ | $58.05_{\pm0.91}$ |
| **tiered-ImageNet** | Conv4 | MAML | $63.26_{\pm0.77}$ | $63.87_{\pm0.77}$ | $64.26_{\pm0.77}$ | $64.52_{\pm0.76}$ | $64.65_{\pm0.76}$ | $47.33_{\pm0.90}$ | $47.74_{\pm0.90}$ | $47.93_{\pm0.91}$ | $48.04_{\pm0.91}$ | $48.08_{\pm0.91}$ |
| | | ours | $64.77_{\pm0.75}$ | $65.56_{\pm0.74}$ | $65.91_{\pm0.74}$ | $66.01_{\pm0.74}$ | $66.05_{\pm0.74}$ | $47.34_{\pm0.88}$ | $47.74_{\pm0.90}$ | $48.07_{\pm0.90}$ | $48.21_{\pm0.90}$ | $48.29_{\pm0.90}$ |
| | Conv6 | MAML | $66.78_{\pm0.79}$ | $67.26_{\pm0.79}$ | $67.40_{\pm0.79}$ | $67.42_{\pm0.79}$ | $67.51_{\pm0.79}$ | $50.40_{\pm0.96}$ | $50.74_{\pm0.95}$ | $50.96_{\pm0.95}$ | $51.00_{\pm0.95}$ | $51.07_{\pm0.96}$ |
| | | ours | $67.60_{\pm0.79}$ | $68.22_{\pm0.78}$ | $68.56_{\pm0.78}$ | $68.77_{\pm0.78}$ | $68.92_{\pm0.78}$ | $50.87_{\pm0.95}$ | $51.18_{\pm0.95}$ | $51.42_{\pm0.96}$ | $51.50_{\pm0.96}$ | $51.62_{\pm0.97}$ |
| | ResNet10 | MAML | $73.03_{\pm0.85}$ | $75.14_{\pm0.79}$ | $75.22_{\pm0.79}$ | $75.19_{\pm0.79}$ | $75.20_{\pm0.80}$ | $57.80_{\pm0.94}$ | $57.95_{\pm0.96}$ | $58.07_{\pm0.95}$ | $58.16_{\pm0.96}$ | $58.22_{\pm0.96}$ |
| | | ours | $75.77_{\pm0.71}$ | $76.29_{\pm0.70}$ | $76.45_{\pm0.70}$ | $76.44_{\pm0.70}$ | $76.43_{\pm0.70}$ | $59.65_{\pm0.96}$ | $59.94_{\pm0.95}$ | $60.22_{\pm0.94}$ | $60.42_{\pm0.93}$ | $60.53_{\pm0.93}$ |
| | ResNet18 | MAML | $73.90_{\pm0.79}$ | $74.88_{\pm0.76}$ | $74.98_{\pm0.77}$ | $74.89_{\pm0.77}$ | $74.94_{\pm0.77}$ | $57.71_{\pm1.00}$ | $57.88_{\pm1.00}$ | $57.90_{\pm1.01}$ | $57.92_{\pm1.01}$ | $57.99_{\pm1.01}$ |
| | | ours | $74.67_{\pm0.74}$ | $74.86_{\pm0.75}$ | $74.96_{\pm0.74}$ | $75.02_{\pm0.74}$ | $75.01_{\pm0.74}$ | $57.80_{\pm0.98}$ | $57.97_{\pm0.98}$ | $58.09_{\pm0.99}$ | $58.14_{\pm0.99}$ | $58.10_{\pm0.98}$ |
| **CIFAR-FS** | Conv4 | MAML | $69.97_{\pm0.75}$ | $70.54_{\pm0.76}$ | $71.04_{\pm0.76}$ | $71.20_{\pm0.76}$ | $71.32_{\pm0.76}$ | $51.84_{\pm0.93}$ | $52.31_{\pm0.93}$ | $52.52_{\pm0.94}$ | $52.74_{\pm0.94}$ | $52.89_{\pm0.94}$ |
| | | ours | $70.32_{\pm0.76}$ | $71.08_{\pm0.77}$ | $71.52_{\pm0.75}$ | $71.68_{\pm0.75}$ | $71.86_{\pm0.76}$ | $52.01_{\pm0.99}$ | $52.46_{\pm0.98}$ | $52.68_{\pm0.98}$ | $52.84_{\pm0.99}$ | $53.02_{\pm0.99}$ |
| | Conv6 | MAML | $74.00_{\pm0.79}$ | $74.56_{\pm0.78}$ | $74.75_{\pm0.77}$ | $74.78_{\pm0.77}$ | $74.80_{\pm0.77}$ | $57.83_{\pm0.95}$ | $58.10_{\pm0.95}$ | $58.29_{\pm0.96}$ | $58.38_{\pm0.96}$ | $58.47_{\pm0.96}$ |
| | | ours | $74.47_{\pm0.78}$ | $74.99_{\pm0.78}$ | $75.24_{\pm0.78}$ | $75.38_{\pm0.78}$ | $75.52_{\pm0.77}$ | $58.12_{\pm0.96}$ | $58.43_{\pm0.96}$ | $58.58_{\pm0.97}$ | $58.70_{\pm0.97}$ | $58.76_{\pm0.97}$ |
| | ResNet10 | MAML | $77.12_{\pm0.77}$ | $78.75_{\pm0.69}$ | $78.77_{\pm0.69}$ | $78.72_{\pm0.70}$ | $78.69_{\pm0.71}$ | $65.42_{\pm0.98}$ | $65.82_{\pm0.99}$ | $65.81_{\pm0.99}$ | $65.98_{\pm0.98}$ | $66.03_{\pm0.97}$ |
| | | ours | $78.90_{\pm0.70}$ | $79.30_{\pm0.68}$ | $79.39_{\pm0.68}$ | $79.34_{\pm0.69}$ | $79.37_{\pm0.69}$ | $66.17_{\pm0.95}$ | $66.04_{\pm0.96}$ | $66.07_{\pm0.97}$ | $66.16_{\pm0.96}$ | $66.24_{\pm0.96}$ |
| | ResNet18 | MAML | $79.59_{\pm0.70}$ | $80.63_{\pm0.66}$ | $80.68_{\pm0.65}$ | $80.60_{\pm0.65}$ | $80.62_{\pm0.65}$ | $67.44_{\pm0.99}$ | $67.73_{\pm0.99}$ | $67.67_{\pm1.01}$ | $67.67_{\pm1.01}$ | $67.66_{\pm1.02}$ |
| | | ours | $80.18_{\pm0.65}$ | $80.37_{\pm0.65}$ | $80.48_{\pm0.64}$ | $80.60_{\pm0.64}$ | $80.71_{\pm0.64}$ | $69.09_{\pm1.03}$ | $69.35_{\pm1.03}$ | $69.49_{\pm1.03}$ | $69.52_{\pm1.03}$ | $69.56_{\pm1.03}$ |
| **FC100** | Conv4 | MAML | $47.61_{\pm0.72}$ | $48.40_{\pm0.71}$ | $48.79_{\pm0.73}$ | $49.14_{\pm0.73}$ | $49.25_{\pm0.73}$ | $36.08_{\pm0.76}$ | $36.45_{\pm0.76}$ | $36.66_{\pm0.77}$ | $36.82_{\pm0.77}$ | $36.89_{\pm0.76}$ |
| | | ours | $48.01_{\pm0.73}$ | $48.59_{\pm0.72}$ | $49.01_{\pm0.72}$ | $49.30_{\pm0.72}$ | $49.52_{\pm0.72}$ | $36.82_{\pm0.76}$ | $37.08_{\pm0.76}$ | $37.24_{\pm0.76}$ | $37.31_{\pm0.76}$ | $37.45_{\pm0.75}$ |
| | Conv6 | MAML | $46.48_{\pm0.70}$ | $47.30_{\pm0.70}$ | $47.48_{\pm0.71}$ | $47.69_{\pm0.71}$ | $47.82_{\pm0.71}$ | $35.64_{\pm0.76}$ | $35.86_{\pm0.76}$ | $35.99_{\pm0.75}$ | $36.04_{\pm0.76}$ | $36.05_{\pm0.75}$ |
| | | ours | $47.45_{\pm0.72}$ | $48.08_{\pm0.73}$ | $48.56_{\pm0.72}$ | $48.77_{\pm0.72}$ | $48.90_{\pm0.72}$ | $36.40_{\pm0.67}$ | $36.51_{\pm0.67}$ | $36.72_{\pm0.67}$ | $36.74_{\pm0.67}$ | $36.80_{\pm0.67}$ |
| | ResNet10 | MAML | $47.03_{\pm0.73}$ | $49.23_{\pm0.71}$ | $49.11_{\pm0.72}$ | $49.06_{\pm0.72}$ | $49.10_{\pm0.71}$ | $36.33_{\pm0.73}$ | $36.32_{\pm0.73}$ | $36.47_{\pm0.73}$ | $36.63_{\pm0.74}$ | $36.76_{\pm0.74}$ |
| | | ours | $50.86_{\pm0.70}$ | $51.08_{\pm0.70}$ | $51.15_{\pm0.71}$ | $51.17_{\pm0.70}$ | $51.16_{\pm0.70}$ | $37.50_{\pm0.73}$ | $37.52_{\pm0.74}$ | $37.36_{\pm0.74}$ | $37.50_{\pm0.74}$ | $37.55_{\pm0.74}$ |
| | ResNet18 | MAML | $48.56_{\pm0.76}$ | $50.25_{\pm0.76}$ | $50.29_{\pm0.76}$ | $50.16_{\pm0.75}$ | $50.18_{\pm0.76}$ | $35.19_{\pm0.70}$ | $35.36_{\pm0.72}$ | $35.44_{\pm0.72}$ | $35.48_{\pm0.72}$ | $35.57_{\pm0.72}$ |
| | | ours | $51.22_{\pm0.74}$ | $51.41_{\pm0.74}$ | $51.55_{\pm0.74}$ | $51.52_{\pm0.74}$ | $51.63_{\pm0.74}$ | $37.02_{\pm0.72}$ | $37.27_{\pm0.72}$ | $37.26_{\pm0.72}$ | $37.37_{\pm0.72}$ | $37.40_{\pm0.72}$ |

# D  Ablating the proposed condition loss function

We introduced in the main paper that computing the condition number as defined in Equation (4) would ignore the distribution of all but two eigenvalues and thus unnecessarily weaken the training signal if directly used as conditioning objective. In this section, we back up this intuition with empirical insights. In detail, we contrast both versions 1) using our loss defined via the variance of the logarithmic eigenvalues of the approximated Hessian as proposed in the main paper in Equation (9) to 2) simply using the logarithmic condition number computed via the maximum and minimum eigenvalues (Table A3). We find that while using the logarithmic condition number does still lead to a significant improvement of adaptation performance especially during the first few steps when compared to its unconstrained counterpart (MAML), it is notably outperformed by our proposed loss using the variance of the eigenvalues.

Table A3: **Ablating the condition loss function.** Reported are the step-wise classification accuracies on the validation set of *mini*ImageNet (Vinyals et al., 2016) for a 5-way 5-shot scenario (Conv6).

| | 5-shot | | | | |
| --- | --- | --- | --- | --- | --- |
| Loss $\mathcal{L}_\kappa$ | step 1↑ | step 2↑ | step 3↑ | step 4↑ | step 5↑ |
| var(log(ev)) | $63.93_{\pm1.76}$ | $68.44_{\pm1.70}$ | $69.15_{\pm1.70}$ | $69.78_{\pm1.69}$ | $69.83_{\pm1.73}$ |
| $\log(\kappa)$ | $55.30_{\pm2.04}$ | $62.03_{\pm1.87}$ | $63.95_{\pm1.82}$ | $64.98_{\pm1.87}$ | $65.36_{\pm1.86}$ |

# E Preconditioning – Number of parameters and performance

As discussed in the main paper, we compare our approach to other recently published methods that aim to achieve better convergence via preconditioning. Table A4 outlines the different parameter update procedures and highlights the additionally introduced parameters of other methods (blue). Note that these parameters are required at both training and inference time, and lead to a significant increase in parameter count ranging from $96\%$ up to $2235\%$. In contrast, our proposed approach does not require any additional parameters to achieve preconditioning and thus allows to use more powerful backbones if increased parameter counts can be tolerated – enabling our method to outperform others across the entire parameter-accuracy spectrum. While we show a visualization outlining the interplay between the total number of parameters and achieved accuracies within the main paper, we provide extended details regarding the explicit parameter counts and accuracy values in Table A5.

Table A4: **Preconditioned parameter updates.** Detailed are the different ways of updating the parameters for recently published preconditioning methods. Additionally introduced parameters are highlighted in blue, and are required at both training and inference time (cf. Table A5).

| Method | Inner Loop Param. Update |
|---|---|
| MAML (Finn et al., 2017) | $\boldsymbol{\theta}_\tau^{(k)} = \boldsymbol{\theta}_\tau^{(k-1)} - \alpha\nabla_{\boldsymbol{\theta}^{(k-1)}}\mathcal{L}(\boldsymbol{\theta}_\tau^{(k-1)})$ |
| Ours | $\boldsymbol{\theta}_\tau^{(k)} = \boldsymbol{\theta}_\tau^{(k-1)} - \alpha\nabla_{\boldsymbol{\theta}^{(k-1)}}\mathcal{L}(\boldsymbol{\theta}_\tau^{(k-1)})$ |
| Meta-SGD (Li et al., 2017) | $\boldsymbol{\theta}_\tau^{(k)} = \boldsymbol{\theta}_\tau^{(k-1)} - \alpha\,\mathrm{diag}(\phi)\nabla_{\boldsymbol{\theta}^{(k-1)}}\mathcal{L}(\boldsymbol{\theta}_\tau^{(k-1)})$ |
| MC (Park & Oliva, 2019) | $\boldsymbol{\theta}_\tau^{(k)} = \boldsymbol{\theta}_\tau^{(k-1)} - \alpha\,M(\boldsymbol{\theta}_\tau^{(k-1)},\psi)\nabla_{\boldsymbol{\theta}^{(k-1)}}\mathcal{L}(\boldsymbol{\theta}_\tau^{(k-1)})$ |
| ModGrad (Simon et al., 2020) | $\boldsymbol{\theta}_\tau^{(k)} = \boldsymbol{\theta}_\tau^{(k-1)} - \alpha\,M_\tau^{(k-1)}(\Psi)\nabla_{\boldsymbol{\theta}^{(k-1)}}\mathcal{L}(\boldsymbol{\theta}_\tau^{(k-1)})$ |
| Warp-MAML (Flennerhag et al., 2019) | $\boldsymbol{\theta}_\tau^{(k)} = \boldsymbol{\theta}_\tau^{(k-1)} - \alpha\nabla_{\boldsymbol{\theta}^{(k-1)}}\mathcal{L}(\boldsymbol{\theta}_\tau^{(k-1)},\zeta)$ |

Table A5: **Preconditioning methods, number of parameters and accuracies.** Obtained for 5-way 5-shot evaluated on the *mini*ImageNet test set. Reported are results for MAML (Finn et al., 2017), Meta-SGD (Li et al., 2017), MC (Park & Oliva, 2019), ModGrad (Simon et al., 2020) and Warp-MAML (Flennerhag et al., 2019). [†]denotes reimplemented versions (cf. Table 1, main paper).

| Method | Backbone Architecture | Parameter increase↓ | Test Accuracy | Rel. Acc. increase↑ | #Total Parameters | #Backbone Parameters |
|---|---|---|---|---|---|---|
| MAML | Conv4 (32) | – | $63.11_{\pm0.92}$ | – | 32,901 | 32,901 |
| ours | Conv4 (32) | $+\ \ \ \mathbf{0\%}$ | $63.33_{\pm0.72}$ | $\mathbf{+0.3\%}$ | 32,901 | 32,901 |
| Meta-SGD | Conv4 (32) | $\mathbf{+100\%}$ | $64.03_{\pm0.94}$ | $\mathbf{+1.5\%}$ | 65,802 | 32,901 |
| ours | Conv6 (32) | $+\ \ \mathbf{57\%}$ | $64.47_{\pm0.71}$ | $\mathbf{+2.2\%}$ | 51,525 | 51,525 |
| MAML[†] | Conv4 (64) | – | $64.50_{\pm0.69}$ | – | 121,093 | 121,093 |
| ours | Conv4 (64) | $+\ \ \ \mathbf{0\%}$ | $65.26_{\pm0.67}$ | $\mathbf{+\ 1.2\%}$ | 121,093 | 121,093 |
| ModGrad | Conv4 (64) | $\mathbf{+873\%}$ | $69.17_{\pm0.69}$ | $\mathbf{+\ 7.2\%}$ | 1,178,019 | 121,093 |
| ours | Conv6 (64) | $+\ \ \mathbf{61\%}$ | $68.43_{\pm0.71}$ | $\mathbf{+\ 6.1\%}$ | 195,205 | 195,205 |
| ours | Conv6 (128) | $\mathbf{+527\%}$ | $71.00_{\pm0.68}$ | $\mathbf{+10.1\%}$ | 759,045 | 759,045 |
| MAML[†] | Conv4 (128) | – | $66.06_{\pm0.71}$ | – | 463,365 | 463,365 |
| ours | Conv4 (128) | $+\ \ \ \mathbf{0\%}$ | $68.07_{\pm0.70}$ | $\mathbf{+3.0\%}$ | 463,365 | 463,365 |
| Warp-MAML | Conv4 (128) | $+\ \ \mathbf{96\%}$ | $68.4\ _{\pm0.92}$ | $\mathbf{+3.5\%}$ | 906,885 | 463,365 |
| MC | Conv4 (128) | $\mathbf{+2235\%}$ | $68.01_{\pm0.73}$ | $\mathbf{+3.0\%}$ | 10,818,928 | 463,365 |
| ours | Conv6 (128) | $+\ \ \mathbf{64\%}$ | $71.00_{\pm0.68}$ | $\mathbf{+7.5\%}$ | 759,045 | 759,045 |

# F Algorithm for better conditioned meta-learning

Algorithm 1 shows the concise form of how the conditioning loss presented in the main paper is used in the context of gradient-based few-shot meta-learning. The algorithm follows the concept introduced by Finn et al. (2017) for MAML, with the addition of using our reformulated problem setting and in this way computing the condition information for each stage of the parameters updated during the inner loop (Lines 8 and 9). The outer loop then incorporates the conditioning constraint

(Line 12) as introduced in Equations (9) and (10) of the main paper and computes the overall task loss (Line 13). After completing all tasks in the current task batch, the network's parameters are then updated (Line 15) by considering both the classification and condition objectives, encouraging the model to learn a well-conditioned parameter space while solving the classification challenge.

---

**Algorithm 1** Learning a Better Conditioned Parameter Space

---

**Require:** $p(\mathcal{T})$; $\alpha, \beta, \gamma$          $\triangleright$ Distribution over tasks; Hyperparameters
1:   $\boldsymbol{\theta}^* \leftarrow$ Random initialization
2: **while** not done **do**
3:     $\{\tau_1, \ldots, \tau_B\} \sim p(\mathcal{T})$          $\triangleright$ Sample a batch of tasks
4:     **for all** $\tau_i$ **do**
5:        $\boldsymbol{\theta}_{\tau_i}^0 \leftarrow \boldsymbol{\theta}^*$
6:        $(\mathcal{D}_{\tau_i}^{\text{train}}, \mathcal{D}_{\tau_i}^{\text{val}}) \sim \tau_i$          $\triangleright$ Sample train and validation set
7:        **for** $k$ in $\{1, \ldots, K\}$ inner-loop update steps **do**          $\triangleright$ Inner-loop adaptation
8:           Compute $\mathbf{J}^{(k)}$ via $\mathcal{L}(\mathcal{D}_{\tau_i}^{\text{train}}, \boldsymbol{\theta}_{\tau_i}^{(k-1)})$          $\triangleright$ Following Equations (5) - (8)
9:           Compute and temporarily store $\boldsymbol{\lambda}(\mathbf{J}^{(k)}\mathbf{J}^{(k)\top})$      $\triangleright$ Eigenvalues of approx. Hessian
10:          $\boldsymbol{\theta}_{\tau_i}^{(k)} \leftarrow \boldsymbol{\theta}_{\tau_i}^{(k-1)} - \alpha \nabla_{\boldsymbol{\theta}_{\tau_i}^{(k-1)}} \mathcal{L}(\mathcal{D}_{\tau_i}^{\text{train}}, \boldsymbol{\theta}_{\tau_i}^{(k-1)})$          $\triangleright$ Inner-loop parameter update
11:        **end for**
12:        $\mathcal{L}_\kappa\left(\boldsymbol{\theta}_{\tau_i}^{(K)}(\mathcal{D}_{\tau_i}^{\text{train}}, \boldsymbol{\theta}^*)\right) = \frac{1}{K}\sum_{k=1}^{K} \text{Var}\left(\log_{10}\left(\boldsymbol{\lambda}\left(\mathbf{J}^{(k)}\mathbf{J}^{(k)\top}\right)\right)\right)$          $\triangleright$ Cond. loss
13:        $\mathcal{L}_{\tau_i} = \mathcal{L}\left(\mathcal{D}_{\tau_i}^{\text{val}}, \ \boldsymbol{\theta}_{\tau_i}^{(K)}(\mathcal{D}_{\tau_i}^{\text{train}}, \boldsymbol{\theta}^*)\right) + \gamma \mathcal{L}_\kappa\left(\boldsymbol{\theta}_{\tau_i}^{(K)}(\mathcal{D}_{\tau_i}^{\text{train}}, \boldsymbol{\theta}^*)\right)$ $\triangleright$ Overall task loss
14:     **end for**
15:     $\boldsymbol{\theta}^* \leftarrow \boldsymbol{\theta}^* - \beta \nabla_{\boldsymbol{\theta}^*} \sum_{i=1}^{B} \mathcal{L}_{\tau_i}$          $\triangleright$ Meta update overall parameter set
16: **end while**

---

## G   Details on many-way multi-shot scenarios

A detailed version of the results used for the visualization of different 5-way $K$-shot and $N$-way 5-shot scenarios depicted in the main paper are presented in Table A6, including the 95% confidence intervals. While enforcing a well-conditioned parameter space for the inner-loop optimization leads to significantly better first-step adaptation results, it can also be observed that the conditioning seems to additionally improve the overall results achieved after 5 updates. The results further indicate that the adaptation of the baseline parameters during the initial steps (mainly 1-3) differs dependent on the number of shots, and seems to be increasingly delayed to the last steps for settings with a higher number of shots (e.g., 42.10% vs. 21.14% vs. 20.32% after 3 updates for $k = 10$, $k = 15$ and $k = 20$, respectively).

## H   Constraining parameter subsets

As discussed in the main paper, we choose to apply our proposed conditioning constraint only to a subset of the network's parameters to increase efficiency and scalability. We demonstrated that the development of the condition number calculated with respect to only the parameters of the classifier is representative for the condition number calculated with respect to the full set of network parameters. In this section, we provide the visualisations of the development of all evaluated subsets. It is to be noted that for all depicted results, the condition constraint is enforced to the parameter subset denoted in the respective legend. As can be observed in Figure A1, all subsets except for the batchnorm of the embedding layer '*eBN*' demonstrate a development of the condition number that is very similar to the one of the condition number *w.r.t.* the full parameter set. For completeness, we additionally provide the development of the condition number *w.r.t.* the full parameter set if the model is trained *without* our proposed conditioning loss in Figure A1 (h) (i.e., conventional MAML baseline like proposed by Finn et al. (2017)) – demonstrating the significantly higher condition number and thus

Table A6: **Many-way multi-shot experiments.** Average test accuracy for various 5-way $K$-shot and $N$-way 5-shot scenarios evaluated on the *mini*ImageNet (Vinyals et al., 2016) test set using a Conv6 architecture.

| | Setting | Method | step 1↑ | step 2↑ | step 3↑ | step 4↑ | step 5↑ |
|---|---|---|---|---|---|---|---|
| 5-way | 1-shot | MAML | $21.40_{\pm0.23}$ | $30.85_{\pm0.66}$ | $45.91_{\pm0.87}$ | $50.30_{\pm0.88}$ | $51.22_{\pm0.88}$ |
| | | ours | $50.92_{\pm0.85}$ | $52.98_{\pm0.89}$ | $53.18_{\pm0.89}$ | $53.28_{\pm0.89}$ | $53.34_{\pm0.89}$ |
| | 5-shot | MAML | $20.19_{\pm0.07}$ | $24.17_{\pm0.39}$ | $57.25_{\pm0.72}$ | $64.24_{\pm0.73}$ | $65.96_{\pm0.71}$ |
| | | ours | $62.31_{\pm0.72}$ | $66.66_{\pm0.71}$ | $67.63_{\pm0.72}$ | $68.21_{\pm0.71}$ | $68.43_{\pm0.71}$ |
| | 10-shot | MAML | $20.00_{\pm0.00}$ | $20.07_{\pm0.04}$ | $42.10_{\pm0.70}$ | $65.48_{\pm0.74}$ | $70.66_{\pm0.67}$ |
| | | ours | $64.82_{\pm0.72}$ | $70.35_{\pm0.70}$ | $71.65_{\pm0.67}$ | $72.68_{\pm0.68}$ | $73.13_{\pm0.67}$ |
| | 15-shot | MAML | $20.00_{\pm0.00}$ | $20.00_{\pm0.00}$ | $21.14_{\pm0.18}$ | $68.00_{\pm0.67}$ | $71.22_{\pm0.63}$ |
| | | ours | $63.91_{\pm0.69}$ | $70.18_{\pm0.65}$ | $72.30_{\pm0.65}$ | $73.58_{\pm0.63}$ | $74.01_{\pm0.63}$ |
| | 20-shot | MAML | $20.00_{\pm0.00}$ | $20.00_{\pm0.00}$ | $20.32_{\pm0.08}$ | $69.41_{\pm0.65}$ | $72.98_{\pm0.61}$ |
| | | ours | $64.83_{\pm0.66}$ | $71.56_{\pm0.63}$ | $73.56_{\pm0.60}$ | $74.85_{\pm0.60}$ | $75.55_{\pm0.58}$ |
| 5-shot | 5-way | MAML | $20.19_{\pm0.07}$ | $24.17_{\pm0.39}$ | $57.25_{\pm0.72}$ | $64.24_{\pm0.73}$ | $65.96_{\pm0.71}$ |
| | | ours | $62.31_{\pm0.72}$ | $66.66_{\pm0.71}$ | $67.63_{\pm0.72}$ | $68.21_{\pm0.71}$ | $68.43_{\pm0.71}$ |
| | 10-way | MAML | $10.00_{\pm0.00}$ | $14.11_{\pm0.30}$ | $41.50_{\pm0.43}$ | $47.38_{\pm0.47}$ | $49.59_{\pm0.46}$ |
| | | ours | $42.20_{\pm0.42}$ | $48.94_{\pm0.45}$ | $50.79_{\pm0.46}$ | $51.67_{\pm0.46}$ | $52.02_{\pm0.47}$ |
| | 15-way | MAML | $6.67_{\pm0.00}$ | $7.66_{\pm0.27}$ | $35.32_{\pm0.32}$ | $38.29_{\pm0.31}$ | $41.08_{\pm0.31}$ |
| | | ours | $32.55_{\pm0.30}$ | $39.81_{\pm0.32}$ | $41.83_{\pm0.32}$ | $42.84_{\pm0.32}$ | $43.30_{\pm0.21}$ |

worse-conditioned parameter space that is learned by the unconstrained method. In stark contrast, it can further be observed that the trajectories of the methods actively enforcing conditioning are very close for all subsets where the parameters of the classifier '*cls*' are involved in the conditioning constraint, and that the condition numbers of the actual network ('*all*') is particularly low for all these cases, justifying our choice of using the '*cls*' subset throughout all major experiments in the main paper.

# I Condition number and few-step performance

As discussed in the main paper, the development of the condition number and the validation accuracy are directly related. While we presented the validation accuracies for a Conv4 and Conv6 architecture together with the condition number of inner-loop update step 1 in the main paper, we herein show the detailed development of all five inner-loop update steps. The corresponding visualisations of the classification accuracy achieved on the validation set during training are presented in Figure A2 for a Conv4 and Conv6 architecture trained *without* ('MAML') and *with* ('ours') the proposed conditioning constraint enforced via $\mathcal{L}_\kappa$. We further show the development of the condition number with respect to the parameters of the classifier using the support sets of the training data (left column, $\kappa(\boldsymbol{\theta}_{\text{train}}^{(k)})$) and the validation data (right column, $\kappa(\boldsymbol{\theta}_{\text{valid}}^{(k)})$) in Figure A3 for steps $k = 0$ up to $k = 4$, i.e., all parameter sets that will be updated during the course of the 5 inner-loop update steps. Note that the condition property of the initial parameter space at step 0 is important to perform the first inner-loop update (step 1), which is why we investigate the condition numbers of the parameter sets before each update (i.e., sets at stages 0 - 4 for update steps 1 - 5). Both architectures have been trained on the *tiered*ImageNet dataset (Ren et al., 2018).

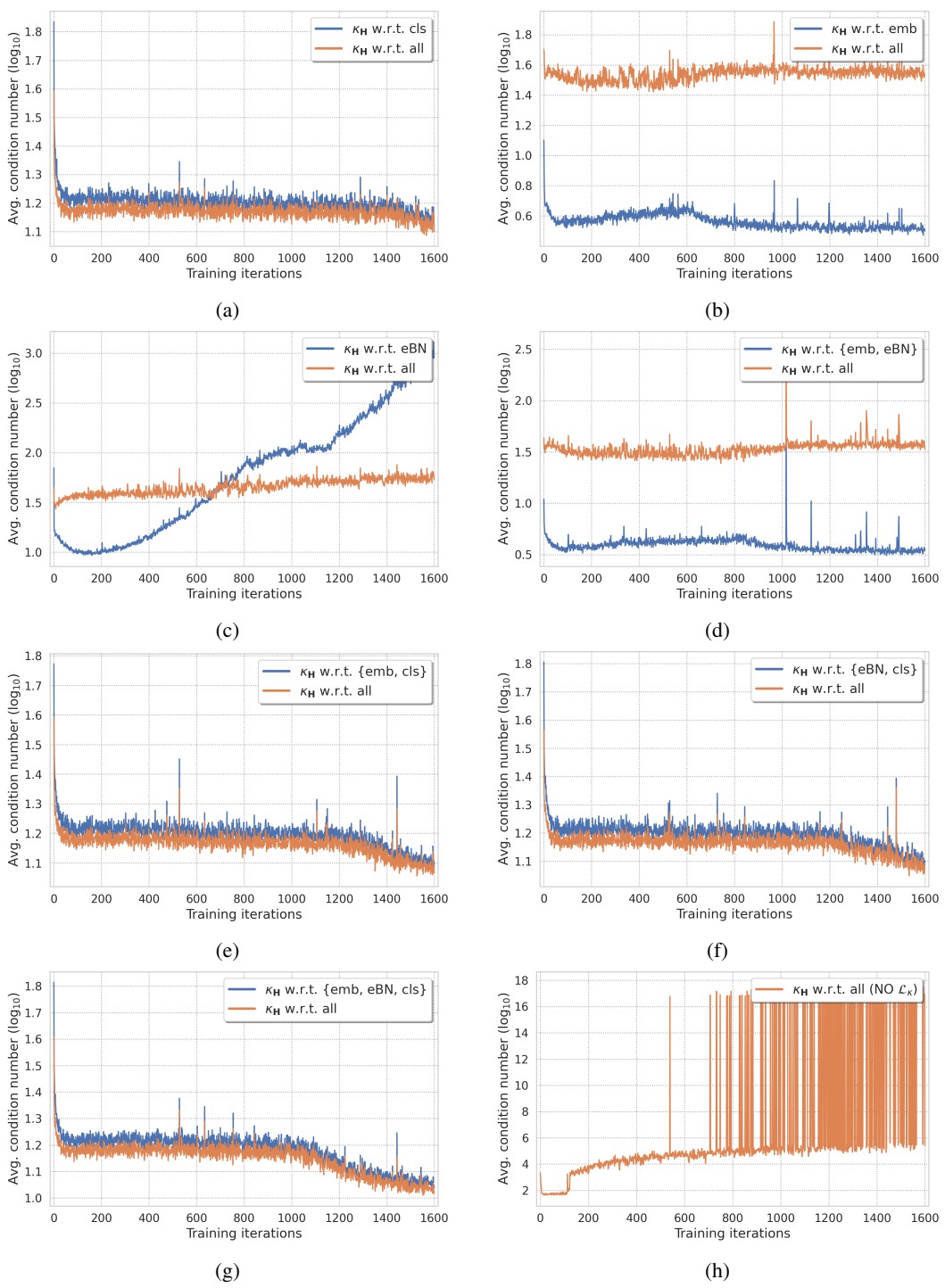

Figure A1: **Constraining a reduced parameter set.** Condition number with respect to the reduced parameter subset denoted in the respective legend, and to all parameters of the model over 1600 iterations on the *mini*ImageNet dataset with a Conv4 architecture for a 5-way 5-shot scenario. Models in (a) - (g) are trained with $\mathcal{L}_\kappa$ *w.r.t.* the respective subset, while (h) shows the development for the model trained without the use of the proposed $\mathcal{L}_\kappa$.

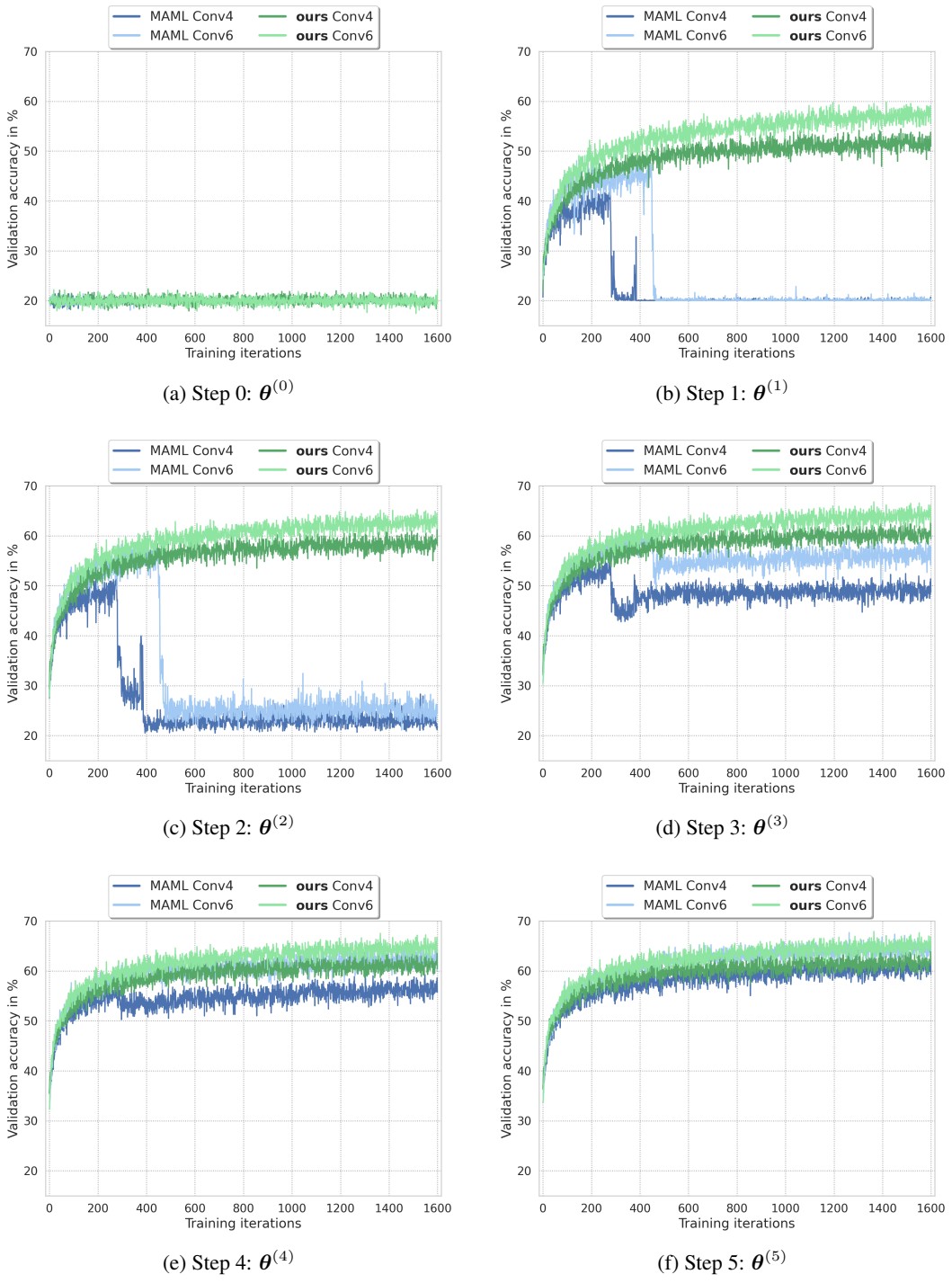

Figure A2: **Validation accuracy over inner-loop update steps.** Reported results obtained by training the baseline *without* ('MAML') and *with* our proposed conditioning constraint ('ours') with respect to the parameters of the model's classifier. Training has been conducted over 1600 iterations in a 5-way 5-shot scenario on the *tiered*ImageNet dataset with a Conv4 and Conv6 architecture.

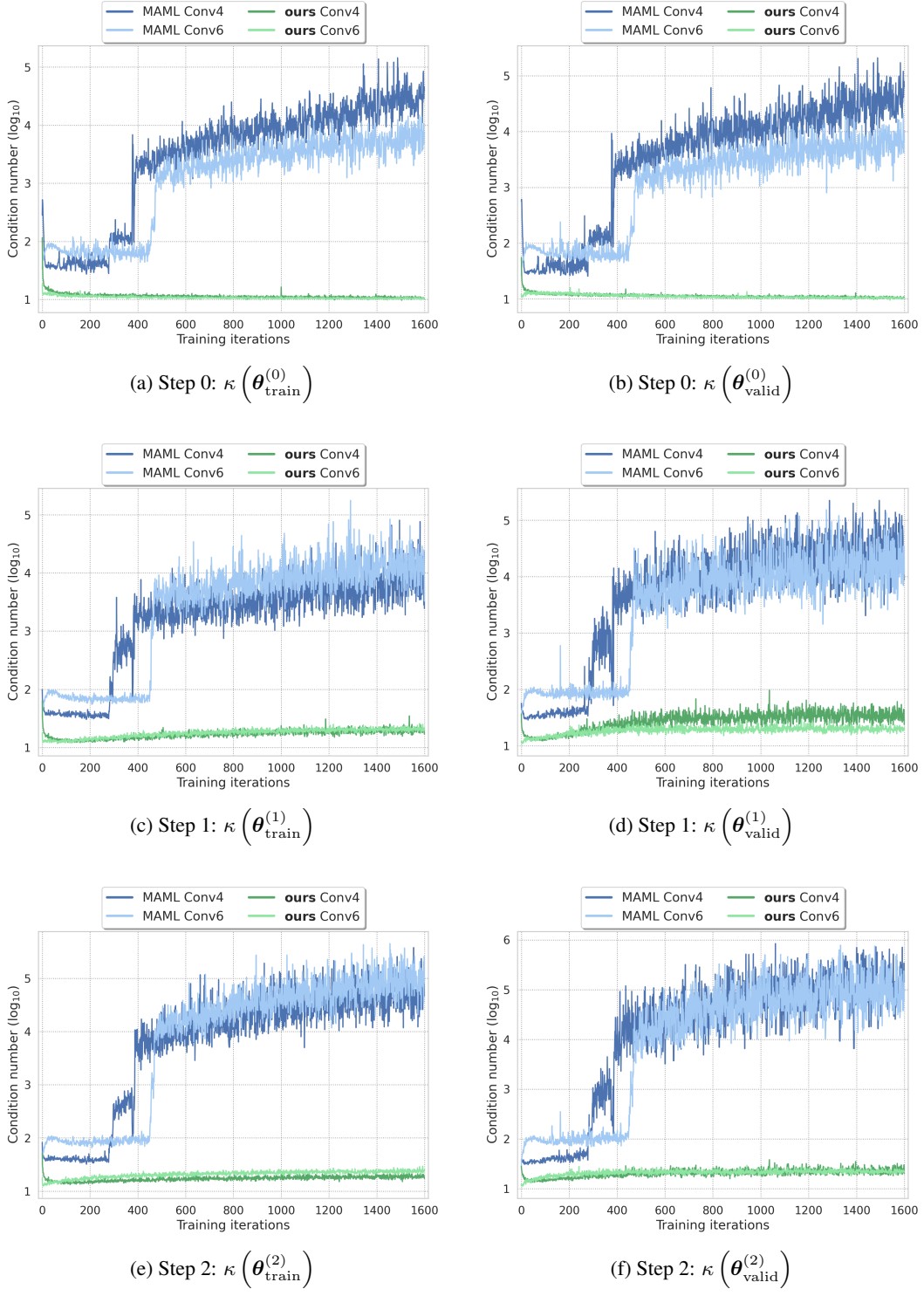

(a) Step 0: $\kappa\left(\boldsymbol{\theta}_{\mathrm{train}}^{(0)}\right)$

(b) Step 0: $\kappa\left(\boldsymbol{\theta}_{\mathrm{valid}}^{(0)}\right)$

(c) Step 1: $\kappa\left(\boldsymbol{\theta}_{\mathrm{train}}^{(1)}\right)$

(d) Step 1: $\kappa\left(\boldsymbol{\theta}_{\mathrm{valid}}^{(1)}\right)$

(e) Step 2: $\kappa\left(\boldsymbol{\theta}_{\mathrm{train}}^{(2)}\right)$

(f) Step 2: $\kappa\left(\boldsymbol{\theta}_{\mathrm{valid}}^{(2)}\right)$

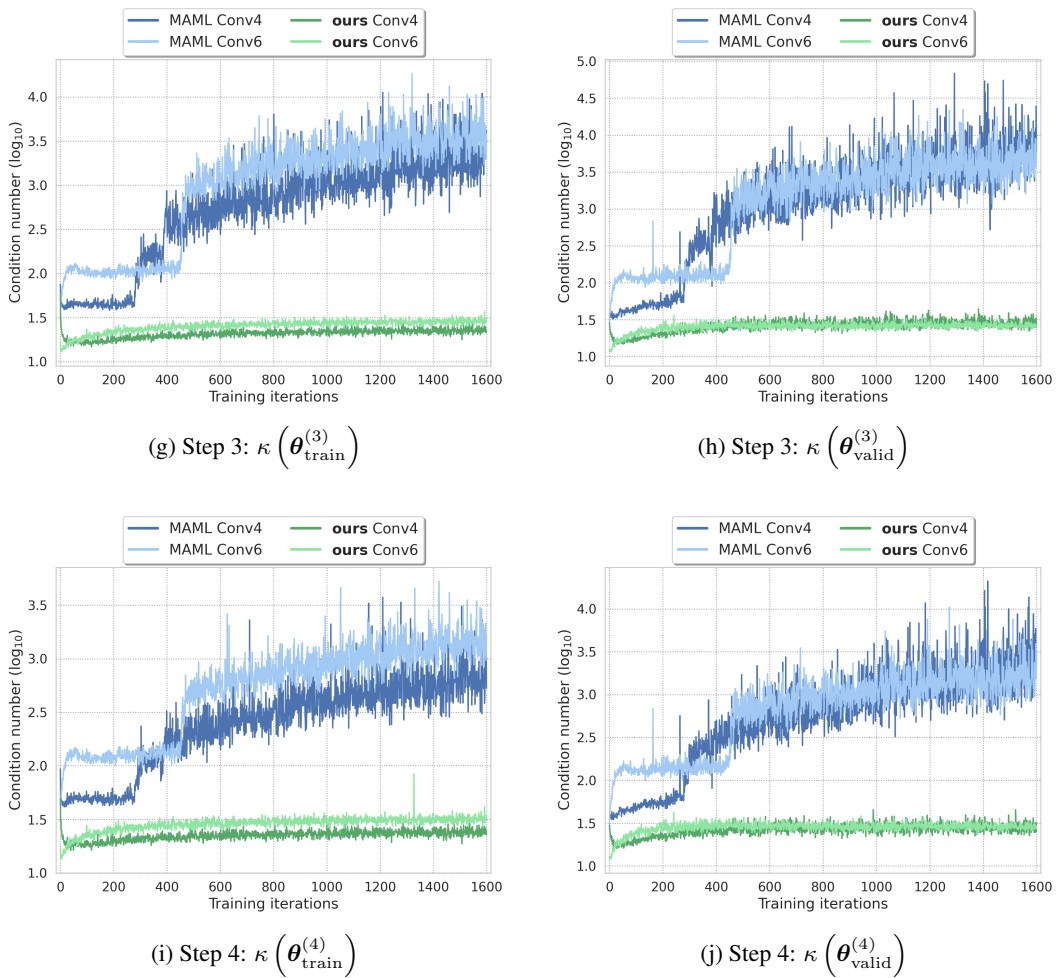

Figure A3: **Condition numbers over inner-loop update steps.** Reported results were obtained by training the baseline *without* ('MAML') and *with* our proposed conditioning constraint ('ours') with respect to the parameters of the model's classifier. Training has been conducted over 1600 iterations in a 5-way 5-shot scenario on the *tiered*ImageNet dataset with a Conv4 and Conv6 architecture. For each update step, we report the condition number computed via either the support set of the training data $\kappa(\boldsymbol{\theta}_{\text{train}}^{(k)})$ or validation data $\kappa(\boldsymbol{\theta}_{\text{valid}}^{(k)})$.