# OpenReview forum: "On Enforcing Better Conditioned Meta-Learning for Rapid Few-Shot Adaptation"
_NeurIPS.cc/2022/Conference — NeurIPS 2022 Accept_

### Official Review · Reviewer_XM8W · 2022-07-09

**Rating:** 6
**Confidence:** 4
**Soundness:** 4 excellent
**Presentation:** 3 good
**Contribution:** 3 good

**Summary:**

Improving the performance of a deep neural network on a new, unseen task with a limited number of new datapoints and adaptation epochs is one of the central problems in modern deep learning. The authors propose to improve the performance of MAML, a benchmark algorithm for few-shot adaptation problems, by pre-conditioning the parameter space using the condition number of the network. Instead of directly using the condition number to condition the bi-level optimization problem of MAML, the authors propose to consider the distribution of all eigenvalues using the approximated logarithmic eigenvalues for increased expressiveness. This simple modification to MAML is shown to be highly effective at improving the few-shot adaptation process across diverse datasets.

**Questions:**

- Can the proposed method also improve MAML's performance in reinforcement learning setting? Experimentally demonstrating the applicability of the proposed method across different problem set-ups or tasks would strengthen the paper's contribution.
- The authors mention that "computing the condition number by using the max and min eigenvalues... would unnecessarily weaken the training signal" (section 3.2). Does using the condition number as opposed to Hessian's logarithmic eigenvalues actually lead to a meaningful difference in adaptation performance?
- Some of the adaptation works have found that only adapting bath norm parameters, instead of the whole network parameters, to reduce the computational cost can be sufficient to improve the adaptation performance. In that regard, it is interesting that adapting cls+emb+eBN parameters performs worse than adapting cls+emb parameters (Table 2), any insight on why additionally adapting bn parameters leads to worse performance?

**Limitations:**

The authors adequately acknowledge the computational complexity of their method in section 5.
Please refer to weakness and questions for additional concerns and questions.

**Strengths And Weaknesses:**

Strength
- The proposed method is theoretically sound, interesting, and novel. Although it is a simple modification to an existing algorithm (MAML), there is enough novelty to be recognized in the idea to better-condition a parameter space using the condition number of the Hessian matrix. In addition to the strong theoretical motivation behind the proposed approach, the authors experimentally demonstrate the relationship between the condition number of a network and its adaptation capabilities.
- The proposed method is shown to be highly effective at allowing more rapid few-shot adaptation. According to the experimental results, the proposed method improves adaptation performance of MAML across all adaptation steps, but the degree of improvement is particularly noteworthy under a limited number of adaptation steps (1 or 2).
- The writing is concise and clear. It was easy to follow how the modified objective function is derived by introducing the conditioning constraint to the MAML bi-level optimization problem.

Weakness
- Maybe more baseline approaches from meta-learning and/or few-shot learning, other than MAML, could be added for comparison. Can MAML + the conditioned parameter space (proposed method) outperform more recent meta-learning algorithms? Also, correct me if I'm wrong, but it appears that the parameter space conditioning can be integrated with other meta-learning algorithms as well. Does it lead to performance improvement regardless of the choice of base meta-learning algorithm?

---

> ### Author Response · Authors · 2022-08-02
> **Response to Reviewer XM8W Part 1**
>
> We thank you for your review and feedback.
>
> >_Maybe more baseline approaches from meta-learning and/or few-shot learning, other than MAML, could be
> added for comparison. Can MAML + the conditioned parameter space (proposed method) outperform more
> recent meta-learning algorithms?_
>
> We provide results comparing to other preconditioning methods using a Conv4 backbone in the table below, and indicate the use of additional parameters that require optimization across all methods. Please note that the goal of work is to provide insights into the benefits of a well- conditioned parameter space and the role of the condition number, rather than optimizing w.r.t. to outperforming other more complex and parameter-intense methods:
>
> | Method | Inner Loop Parameter Update|  Add. Params | Test Acc. |
> |:----|:----|:----:|:--:|
> | MAML [A] |  $\boldsymbol{\theta}^{(k)}\_{\tau} = \boldsymbol{\theta}^{(k-1)}\_{\tau} - \alpha \nabla\_{\boldsymbol{\theta}^{(k-1)}}\mathcal{L}\left(\boldsymbol{\theta}^{(k-1)}\_{\tau}\right) $ | -- | $63.1\pm0.9$ |
> | Ours |    $\boldsymbol{\theta}^{(k)}\_{\tau} = \boldsymbol{\theta}^{(k-1)}\_{\tau} - \alpha \nabla\_{\boldsymbol{\theta}^{(k-1)}}\mathcal{L}\left(\boldsymbol{\theta}^{(k-1)}\_{\tau}\right) $ | -- | $65.3\pm0.7$|
> | Meta-SGD [B]  | $\boldsymbol{\theta}^{(k)}\_{\tau} = \boldsymbol{\theta}^{(k-1)}\_{\tau} - \alpha  \mathrm{diag}(\phi) \nabla\_{\boldsymbol{\theta}^{(k-1)}}\mathcal{L}(\boldsymbol{\theta}^{(k-1)}\_{\tau})$ | $\mathrm{diag}(\phi)$ | $64.0±0.9$ |
> | MC [C]|$\boldsymbol{\theta}^{(k)}\_{\tau} = \boldsymbol{\theta}^{(k-1)}\_{\tau} - \alpha M(\boldsymbol{\theta}^{(k-1)}\_{\tau},\psi) \nabla\_{\boldsymbol{\theta}^{(k-1)}}\mathcal{L}(\boldsymbol{\theta}^{(k-1)}\_{\tau})$ |$\psi$ | $68.0\pm0.7$ |
> | ModGrad [D]|$\boldsymbol{\theta}^{(k)}\_{\tau} = \boldsymbol{\theta}^{(k-1)}\_{\tau} - \alpha M\_{\tau}^{(k-1)}(\Psi) \nabla\_{\boldsymbol{\theta}^{(k-1)}}\mathcal{L}(\boldsymbol{\theta}^{(k-1)}\_{\tau})$ | $\Psi$ | $69.2\pm0.7$ |
> | Warp-MAML [E]|$\boldsymbol{\theta}^{(k)}\_{\tau} = \boldsymbol{\theta}^{(k-1)}\_{\tau} - \alpha \nabla\_{\boldsymbol{\theta}^{(k-1)}}\mathcal{L}(\boldsymbol{\theta}^{(k-1)}\_{\tau}, \zeta)$|$\zeta$ | $68.4\pm0.6$|
>
> [A] Finn et al.: Model-agnostic meta-learning for fast adaptation of deep networks. ICML, 2017
>
> [B] Li et al.: Meta-SGD: Learning to Learn Quickly for Few Shot Learning. arXiv:1707.09835, 2017
>
> [C] Park et al.: Meta-Curvature. NeurIPS, 2019
>
> [D] Simon et al.: On Modulating the Gradient for Meta-Learning. ECCV, 2020
>
> [E] Flennerhag et al.: Meta-Learning with Warped Gradient Descent. ICLR, 2020
>
> >_Also, correct me if I’m wrong, but it appears that the parameter space conditioning can be integrated with other meta-learning algorithms as well. Does it lead to performance improvement regardless of the choice of base meta-learning algorithm?_
>
> While this was beyond the scope of our paper, we agree that our approach should in principle be applicable to other meta-learning algorithms as well. We chose MAML in this specific setting due to its popularity and seminal character, as well as simplicity which allowed us to clearly investigate the contribution and effects of our imposed constraints without interference of additional modules. We would welcome further investigations of our proposed and other previous preconditioning methods within further meta-learning algorithms.
>
> >_Can the proposed method also improve MAML’s performance in reinforcement learning setting? Experimentally demonstrating the applicability of the proposed method across different problem set-ups or tasks would strengthen the paper’s contribution._
>
> Including our proposed approach into a reinforcement learning setting is an interesting suggestion. It is worth considering that reinforcement learning has a *much* noisier (and weaker) training signal than conventional meta learning applied to classification (like the one used within this work). We suspect that it might take longer to realize gains from reducing the condition number on a batch (which is a much noisier estimate of the true condition number) and see this as interesting future work.

---

> > ### Author Response · Authors · 2022-08-02
> > **Response to Reviewer XM8W Part 2**
> >
> > ... continued:
> >
> > >_The authors mention that "computing the condition number by using the max and
> > min eigenvalues... would unnecessarily weaken the training signal" (section 3.2). Does using the condition
> > number as opposed to Hessian’s logarithmic eigenvalues actually lead to a meaningful difference in adaptation
> > performance?_
> >
> > Using the actual condition number, or rather the more stable version log(cond-nmbr), does lead to a meaningful difference in adaptation performance, however takes much longer to converge and is outperformed by our proposed loss using the variance of the eigenvalues. Results below show the validation accuracy achieved after each update step after training a Conv6 architecture on miniImageNet in a 5-way 5-shot setting:
> > | Conv6 Val Acc. | step1 | step2 |step3 |step4 |step5 |
> > |:----|:----:|------:|:----:|------:|:----:|
> > | var(log(ev))| 63.93±1.76| 68.44±1.70| 69.15±1.70| 69.78±1.69| 69.83±1.73|
> > | log(cond-nmbr) |55.30±2.04 |62.03±1.87 |63.95±1.82| 64.98±1.87| 65.36±1.86|
> >
> > > _Some of the adaptation works have found that only adapting bath norm parameters,
> > instead of the whole network parameters, to reduce the computational cost can be sufficient to improve the
> > adaptation performance. In that regard, it is interesting that adapting cls+emb+eBN parameters performs
> > worse than adapting cls+emb parameters (Table 2), any insight on why additionally adapting bn parameters
> > leads to worse performance?_
> >
> > One reason that other authors limit their focus on adapting only the the batchnorm
> > parameters is that this significantly reduces the dimensionality of the update space, which prevents
> > overfitting and can tend to also reduce the condition number for these updates. One possible intuition
> > why we observe that conditioning on the cls+emb parameter set achieves better performance than
> > on the cls+emb+eBN set is that by adding the additional parameters, we actually increase the
> > dimensionality of our condition-enforced update space and might thus provoke other negative
> > impacts. Our investigations presented in the supplementary material in Figure A1 also show that
> > enforcing the conditioning constraint on only the batch norm parameters does not prove particularly
> > helpful to reduce the overall condition number of the network compared to other sets.

---

> ### Comment · Reviewer_XM8W · 2022-08-09
> **Insightful and thorough comments; keeping my score**
>
> I would like to thank the authors for providing me a strong and detailed author response. Clearly, they have put in a lot of effort and thought into answering the raised questions. After reading the author response, I feel that the core concept of the paper not only is quite neat and well supported with empirically results but also has the potential to inspire interesting future research into fundamentally improving the performance of meta-learning. Therefore, I am keeping my score 6: Weak Accept.

---

### Official Review · Reviewer_bs8R · 2022-07-11

**Rating:** 6
**Confidence:** 3
**Soundness:** 3 good
**Presentation:** 3 good
**Contribution:** 3 good

**Summary:**

The paper proposes a regularisation term for the outer-loop of MAML to encourage a well conditioned parameter space that improves inner loop adaptation. The experiments suggest that the condition number is correlated to the performance within a few gradient updates. Training with the proposed regularisation term is shown to empirically improve upon MAML.



**Questions:**


It is not clear to me why MAML's training collapses in Figure 2.


Although there has been previous work in preconditioning, this is the first method that I'm aware of that does so without additional parameters. On Page 9 Line 299-300, it is mentioned that existing preconditioning methods "[incur] an often extensive number of additional parameters". This seems to suggest that some methods may not require a lot more parameters. If so, how does the number of parameters required compare to the existing preconditioning methods and how do the performance differ? This would allow a potential user to make a decision between which method to try.

I find it interesting that the proposed method can scale to many more gradient steps (Table 3). How does MAML fare when scaling to those same number of gradient steps at test time? Does MAML continue to perform similarly or better than with 5 gradient steps? I am not convinced of the benefit of choosing the number of adaptation steps at inference steps. For example, at test time, it makes sense to evaluate the model on the same number of inner loop updates as it was trained on. Although, it is beneficial if improvements can be further gained with more training (as we see in Table 3).


On Page 9 Line 316, it's mentioned that the proposed method "significantly [outperforms] unconstrained methods during initial adaptation". However, the only method compared is MAML, so it would be more precise to say "significantly [outperforms] its unconstrained counterpart during initial adaptation"

If the parameter space is well-conditioned, then I would expect that a single large gradient step from the meta-initialization could already train a good model. For example, instead of evaluating the model by fine-tuning with 5 inner loop updates with an inner loop learning rate of 0.1, I'm curious as to the performance of fine-tuning with 1 inner loop update with an inner loop learning rate of say 0.5.

**Limitations:**

Yes

**Strengths And Weaknesses:**


Strengths:
 - Does not require additional parameters for adaptation unlike existing preconditioning methods.
 - Interesting analysis regarding the connection between condition number and few-step performance on an unseen task.
 - Promising experiment results show an empirical benefit of preconditioning.

Weaknesses:
 - Requires higher-order gradients (computationally expensive) like MAML
 - Computing the conditioning term is expensive. As such, instead of applying the conditioning constraint to the entire model, only a small subset of parameters can have the conditioning constraint applied.
 - Missing comparison with existing preconditioning methods (For example, see Section 2.1 of WarpGrad)

Overall, I would rate the paper:

Novelty: Medium

Clarity: High

Significance: Medium

---

> ### Author Response · Authors · 2022-08-02
> **Response to Reviewer bs8R Part1**
>
> We thank you for your review!
>
> >_Computing the conditioning term is expensive. As such, instead of applying the conditioning constraint to the entire model, only a small subset of parameters can have the conditioning constraint applied._
>
> While we agree that computing the condition number w.r.t. all parameters of the network proves expensive, we demonstrate in the paper that enforcing our conditioning loss on a small subset of parameters (e.g. classifier) is sufficient to significantly reduce the condition number
> of the entire network’s parameter space (e.g. Figure 4 of the main paper and Figure A1 of the supplementary material).
>
> >_Missing comparison with existing preconditioning methods (For example, see Section 2.1 of WarpGrad)_
>
> We thank you for raising this point. While we discuss the related preconditioning methods
> within our related work and the introduction, we agree that a concise overview over the differences
> in update rules and parameter sets will prove beneficial to ease comparison between methods. We
> provide an overview of our method in context to existing preconditioning methods in the following
> table. Please note that the goal of this work is to provide insights into the benefits of a well-
> conditioned parameter space for learning-based methods and the role of the condition number,
> which we hope will motivate further research in this area (rather than optimizing towards competing
> with other more complex and parameter-intense methods).
>
> | Method | Inner Loop Parameter Update|  Add. Params | Test Acc. |
> |:----------|:-------------|:------:|:----:|
> | MAML [A] |  $\boldsymbol{\theta}^{(k)}\_{\tau} = \boldsymbol{\theta}^{(k-1)}\_{\tau} - \alpha \nabla\_{\boldsymbol{\theta}^{(k-1)}}\mathcal{L}\left(\boldsymbol{\theta}^{(k-1)}\_{\tau}\right) $ | -- | $63.1\pm0.9$ |
> | Ours |    $\boldsymbol{\theta}^{(k)}\_{\tau} = \boldsymbol{\theta}^{(k-1)}\_{\tau} - \alpha \nabla\_{\boldsymbol{\theta}^{(k-1)}}\mathcal{L}\left(\boldsymbol{\theta}^{(k-1)}\_{\tau}\right) $ | -- | $65.3\pm0.7$|
> | Meta-SGD [B]  | $\boldsymbol{\theta}^{(k)}\_{\tau} = \boldsymbol{\theta}^{(k-1)}\_{\tau} - \alpha  \mathrm{diag}(\phi) \nabla\_{\boldsymbol{\theta}^{(k-1)}}\mathcal{L}(\boldsymbol{\theta}^{(k-1)}\_{\tau})$ | $\mathrm{diag}(\phi)$ | $64.0±0.9$ |
> | MC [C]|$\boldsymbol{\theta}^{(k)}\_{\tau} = \boldsymbol{\theta}^{(k-1)}\_{\tau} - \alpha M(\boldsymbol{\theta}^{(k-1)}\_{\tau},\psi) \nabla\_{\boldsymbol{\theta}^{(k-1)}}\mathcal{L}(\boldsymbol{\theta}^{(k-1)}\_{\tau})$ |$\psi$ | $68.0\pm0.7$ |
> | ModGrad [D]|$\boldsymbol{\theta}^{(k)}\_{\tau} = \boldsymbol{\theta}^{(k-1)}\_{\tau} - \alpha M\_{\tau}^{(k-1)}(\Psi) \nabla\_{\boldsymbol{\theta}^{(k-1)}}\mathcal{L}(\boldsymbol{\theta}^{(k-1)}\_{\tau})$ | $\Psi$ | $69.2\pm0.7$ |
> | Warp-MAML [E]|$\boldsymbol{\theta}^{(k)}\_{\tau} = \boldsymbol{\theta}^{(k-1)}\_{\tau} - \alpha \nabla\_{\boldsymbol{\theta}^{(k-1)}}\mathcal{L}(\boldsymbol{\theta}^{(k-1)}\_{\tau}, \zeta)$|$\zeta$ | $68.4\pm0.6$|
>
> [A] Finn et al.: Model-agnostic meta-learning for fast adaptation of deep networks. ICML, 2017
>
> [B] Li et al.: Meta-SGD: Learning to Learn Quickly for Few Shot Learning. arXiv:1707.09835, 2017
>
> [C] Park et al.: Meta-Curvature. NeurIPS, 2019
>
> [D] Simon et al.: On Modulating the Gradient for Meta-Learning. ECCV, 2020
>
> [E] Flennerhag et al.: Meta-Learning with Warped Gradient Descent. ICLR, 2020
>
> > _It is not clear to me why MAML’s training collapses in Figure 2._
>
> We suspect that MAML is able to find a meta-initialization that overfits to the tasks and essentially learns that it does not need to perform the initial few steps into the ’best’ possible direction to still sufficiently decrease it’s overall loss. By doing so, it is still able to achieve satisfying performance during the last few updates but can no longer reach its highest possible one – as the comparison to our conditioned MAML clearly demonstrates (Table 3). This behavior further shows the instability / inefficiency of the initial steps of the unconstrained counterpart and hints at the more general challenge: it is not clear in advance for any application how many steps and at which step size one should choose. We see our method as first step towards alleviating this usually
> heuristically approached problem.

---

> > ### Author Response · Authors · 2022-08-02
> > **Response to Reviewer bs8R Part2**
> >
> > ... continued
> >
> > >_Although there has been previous work in preconditioning, this is the first method that I’m aware of that does so without additional parameters. On Page 9 Line 299-300, it is mentioned that existing preconditioning methods "[incur] an often extensive number of additional parameters". This seems to suggest that some methods may not require a lot more parameters. If so, how does the number of parameters
> > required compare to the existing preconditioning methods and how do the performance differ? This would allow a potential user to make a decision between which method to try._
> >
> > We thank you for drawing our attention to this formulation, and acknowledge our unfortunate way of wording this part. We agree with you on the fact that to the best of our knowledge, we are the first to show preconditioning without additional parameters (please see Table above). To provide better comparison and outline additionally require parameters of other methods, we are adding a section detailing the difference between other preconditioning methods regarding parameters they require as well as the performance gains they can achieve to the supplementary material of our revised version.
> >
> > >_I find it interesting that the proposed method can scale to many more gradient
> > steps (Table 3). How does MAML fare when scaling to those same number of gradient steps at test time?
> > Does MAML continue to perform similarly or better than with 5 gradient steps? I am not convinced of the
> > benefit of choosing the number of adaptation steps at inference steps. For example, at test time, it makes
> > sense to evaluate the model on the same number of inner loop updates as it was trained on. Although, it is
> > beneficial if improvements can be further gained with more training (as we see in Table 3)._
> >
> > We provide the detailed results regarding further adaptation beyond the training horizon for our method as well as MAML in the supplementary for all combinations of backbones and datasets in Table A2. MAML shows similar behavior in that it is able to further improve its performance when evaluated for more steps, but is generally not able to entirely ’catch up’ with the results of our
> > method. We agree that running an experiment in the same configuration is generally a valid approach for
> > loosely or unconstrained setups (compute, energy, time) and when the method is trained on only
> > few steps; However, if methods could achieve better results via additional update steps (as shown in
> > our work) or when trained for a significantly higher number of steps, it might prove more beneficial
> > to be able to only take a few in case a fast but slightly less accurate answer is required, or spend
> > more time and get a higher accuracy if the situation and constraints permit.
> >
> > >_On Page 9 Line 316, it’s mentioned that the proposed method "significantly
> > [outperforms] unconstrained methods during initial adaptation". However, the only method compared is
> > MAML, so it would be more precise to say "significantly [outperforms] its unconstrained counterpart during
> > initial adaptation"_
> >
> > We agree with you and thank you for pointing this out; We have corrected the wording in
> > our revised version.
> >
> > >_If the parameter space is well-conditioned, then I would expect that a single
> > large gradient step from the meta-initialization could already train a good model. For example, instead of
> > evaluating the model by fine-tuning with 5 inner loop updates with an inner loop learning rate of 0.1, I’m
> > curious as to the performance of fine-tuning with 1 inner loop update with an inner loop learning rate of say
> > 0.5._
> >
> > We thank you for proposing this interesting ablation. We have run additional experiments with our approach and MAML trained on 5 steps while evaluating on one step with scaled-up learning rate. While our approach evaluated on the same setting as used in training still outperforms its 1-step variants, note that our well-conditioned method with 1 step using 5*lr outperforms the 5-step MAML model.
> >
> > | Conv6   |1step - lr|1step - 2*lr|1step - 3*lr|1step - 4*lr|1step - 5*lr| 5steps - lr|
> > |----------|:-----:|:-----:|:-----:|:-----:|:-----:|:-----:|
> > | Ours| 62.31±0.72| 65.67±0.71| 65.87±0.71| 65.72±0.71| **66.54±0.70**| **68.43±0.71**|
> > | MAML| 20.19±0.07| 20.00±0.00| 20.14±0.05| 23.28±0.37| 26.64±0.51| 65.96±0.71|

---

> > > ### Comment · Reviewer_bs8R · 2022-08-06
> > > **Response**
> > >
> > > I would like to thank the authors for the response and clarifications.
> > >
> > > > We suspect that MAML is able to find a meta-initialization that overfits to the tasks and essentially learns that it does not need to perform the initial few steps into the ’best’ possible direction to still sufficiently decrease it’s overall loss. By doing so, it is still able to achieve satisfying performance during the last few updates but can no longer reach its highest possible one – as the comparison to our conditioned MAML clearly demonstrates (Table 3). This behavior further shows the instability / inefficiency of the initial steps of the unconstrained counterpart and hints at the more general challenge: it is not clear in advance for any application how many steps and at which step size one should choose. We see our method as first step towards alleviating this usually heuristically approached problem.
> > >
> > > Has this occurred in previous works? If so, it would be helpful to reference this as I have not seen it happen before.
> > >
> > > I also believe it would help the table that compare with the existing preconditioning methods if an actual value of the number of additional parameters is included since it is unclear, for example, how many parameters $\zeta$ is. For example, Method A requires $\theta$ additional parameters.
> > >
> > > Practically, I'm unsure about the benefit of using this method, but I believe the insights derived from this paper is interesting.

---

> > > > ### Author Response · Authors · 2022-08-08
> > > > **Response to feedback**
> > > >
> > > > Thank you for your continued feedback!
> > > >
> > > > >_Has this occurred in previous works? [...]_
> > > >
> > > > Although there have been several works (e.g. [Y], [Z]) analyzing different aspects of the MAML algorithm, we are to the best of our knowledge the first to investigate the contribution of each individual update step towards the optimization objective / overall performance in more detail.
> > > >
> > > > [Y] Raghu et al.: Rapid learning or feature reuse? Towards understanding the effectiveness of MAML. ICLR, 2019
> > > >
> > > > [Z] Arnold et al.: When MAML Can Adapt Fast and How to Assist When It Cannot. AISTATS, 2021
> > > >
> > > > >_[...] actual value of the number of additional parameters [...]_
> > > >
> > > > We agree that explicit numbers will provide much better insight for the reader. Note that these numbers are dependent on the specific architecture and ’way of incorporating’ the respective method since most approaches (MC, ModGrad and Warp-MAML) provide different possibilities regarding how many gradients shall be modulated/preconditioned.
> > > >
> > > > We provide the arch. chosen for the experimental result reported in the respective paper (Conv4) here, and will include a more comprehensive discussion into our work.
> > > > Assume we split the Conv4 arch. into its four parameter sub-groups (layers) and a classifier as follows: $\boldsymbol{\theta}$={$
> > > > \boldsymbol{\theta}^{l1},
> > > > \boldsymbol{\theta}^{l2},
> > > > \boldsymbol{\theta}^{l3},
> > > > \boldsymbol{\theta}^{l4},
> > > > \boldsymbol{\theta}^{cl}$}. Each of the first 4 sub-groups then contains a set of convolutional weights $\boldsymbol{\theta}^{\mathrm{li}}_{\mathrm{conv}}$ and potentially additional parameters (e.g. bias, BN). **We focus on the parameters of the convolutional operation within the 2nd layer in the following**. We provide a table to give a concise overview over the additional parameters required to precondition this specific subgroup, and the explanations afterwards in more detail:
> > > >
> > > > | Method | Architecture | sub-group params.|Add. params. for precond. sub-group|
> > > > |:----|:----|:------|:----|
> > > > |MAML [A]| Conv4 (64) | $\|\boldsymbol{\theta}^{\mathrm{l2}}_{\mathrm{conv}}\| = 36{,}864\$| -- |
> > > > |ours| Conv4 (64)| $\|\boldsymbol{\theta}^{\mathrm{l2}}_{\mathrm{conv}}\| = 36{,}864$| -- |
> > > > |Meta-SGD [B]| Conv4 (64)| $\|\boldsymbol{\theta}^{\mathrm{l2}}_{\mathrm{conv}}\| = 36{,}864$| $\|\boldsymbol{\phi}^{\mathrm{l2}}_{\mathrm{conv}}\|=36{,}864$|
> > > > |MC [C]| Conv4 (128)| $\|\boldsymbol{\theta}^{\mathrm{l2}}_{\mathrm{conv}}\| = 147{,}456$| $\|\boldsymbol{\psi}^{\mathrm{l2}}_{\mathrm{conv}}\|=32{,}849$ |
> > > > |ModGrad [D]| Conv4 (64)| $\|\boldsymbol{\theta}^{\mathrm{l2}}_{\mathrm{conv}}\| = 36{,}864$| $\|\boldsymbol{\Psi}^{\mathrm{l2}}_{\mathrm{conv}}\|=485{,}760$|
> > > > |Warp-MAML [E]| Conv4 (128)| $\|\boldsymbol{\theta}^{\mathrm{l2}}_{\mathrm{conv}}\| = 147{,}456$| $\|\boldsymbol{\zeta}^{\mathrm{l2}}_{\mathrm{conv}}\|=147{,}456$ |
> > > >
> > > > - **Ours**: **No add. parameters**
> > > >
> > > > - **Meta-SGD**: One add. parameter for each network parameter, i.e. **doubles** the number of parameters.
> > > >
> > > > - **MC**: Three additional matrices for each parameter sub-group: $M_{i}\in\mathbb{R}^{C_{\mathrm{in}} \times C_{in}}$, $M_{o}\in\mathbb{R}^{C_{\mathrm{out}} \times C_{\mathrm{out}}}$, $M_{f}\in\mathbb{R}^{d \times d}$ with $C_{\mathrm{in}}$, $C_{out}$ the number of input, output channels;  $d$ is kernel-dim.
> > > > Example convolutional weights of $\boldsymbol{\theta}^{\mathrm{l2}}$ with 128 filters used in their paper (without bias potential batch-norm params): $C_{\mathrm{in}}=128$, $C_{\mathrm{out}}=128$, $d=9$; **additional $32{,}849$ parameters** ($\|M_{i}\|=16{,}384$, $\|M_{o}\|=16{,}384$ and $\|M_{f}\|=81$).
> > > > - **ModGrad**: Modulation of each parameter sub-group via two sister-networks $(\phi_1,\phi_1)$ (three if bias used), with 2 FC-layers each. Additional parameters for each network $\phi_i$ are $\phi_i^{\mathrm{fc1}}\in \mathbb{R}^{v\times D_{j}}$ and $\phi_i^{\mathrm{fc2}}\in \mathbb{R}^{D_{j}\times (u+uD_{j})}$; $u$ and $v$ are hyperparameters chosen to $5$ and $300$, $D_{j}$ is the dimension dependent on the weights that shall be modulated.
> > > > Example for $\boldsymbol{\theta}^{\mathrm{l2}}$ using Conv4 with 64 filters as in their paper: plus **additional $485{,}760$ parameters** ($|\phi_1^{\mathrm{fc1}}|=57{,}600$, $|\phi_1^{\mathrm{fc2}}|=185{,}280$, $|\phi_2^{\mathrm{fc1}}|=57{,}600$, $|\phi_2^{\mathrm{fc2}}|=185{,}280$).
> > > > - **Warp-MAML**: Modulation of each parameter sub-group via dedicated warp-modules for each, which differ between architectures (e.g. conv vs ResNet). For the example $\boldsymbol{\theta}^{\mathrm{l2}}$ of a Conv4 architecture with 128 filters as used in their paper, they insert an additional conv-warp-module after each convolutional block, resulting in **additional $147{,}456$ parameters**, i.e. **doubles** the number of parameters.
> > > >
> > > > We hope this provides better insight and further outlines the significant difference in the number of parameters used to achieve preconditioning in other works.

---

### Official Review · Reviewer_DdSu · 2022-07-11

**Rating:** 8
**Confidence:** 4
**Soundness:** 4 excellent
**Presentation:** 4 excellent
**Contribution:** 4 excellent

**Summary:**

- The authors propose a regularization method for improving the conditioning of MAML.
- Experiments show the method consistently reduces number of steps for adaption, and improves accuracy

**Questions:**

Is it the real condition number or the variance of logs in Figure 2?

**Limitations:**

Yes

**Strengths And Weaknesses:**

Strengths:
- method seems very effective at reducing number of required MAML steps
- consistently outperforms MAML
- simple additional loss during training, no cost at inference time
- authors study effect of using parameter subset

Weaknesses:
- precondition loss only applied in top layers (as mentioned by authors), but this may not be a big problem, as the top layers are the ones that are adapted the most during meta-testing

---

> ### Author Response · Authors · 2022-08-02
> **Response to Reviewer DdSu**
>
> We thank you for your feedback.
>
> > _precondition loss only applied in top layers (as mentioned by authors), but this may not be a big problem, as
> the top layers are the ones that are adapted the most during meta-testing_
>
> To add to the fact you mention that mostly the top layers are adapted, we also show in Figure 4 of the main paper as well as Figure 1 of the supplementary material that the condition number of an appropriately chosen parameter subset is a very good predictor for the condition number of the overall network, and applying the loss to the subset does indeed reduce the entire network’s condition number.
>
> >_Is it the real condition number or the variance of logs in Figure 2?_
>
> In Figure 2(c), we show the progress of the ‘real’ condition number of the Hessian w.r.t. all
> parameters of the network, computed via the ratio of maximum and minimum absolute eigenvalue
> as defined in equation (4) using our approximation of the Hessian defined in equation (8). We will
> reformulate the figure’s caption to improve clarity of this fact in our revised version.

---

### Official Review · Reviewer_FA57 · 2022-07-12

**Rating:** 5
**Confidence:** 3
**Soundness:** 2 fair
**Presentation:** 3 good
**Contribution:** 2 fair

**Summary:**

This paper presents the idea of enforcing better condition numbers for inner-loops in meta-learning frameworks. They reformulated meta-learning loss as the least-square formulation, which enables them to easily approximate condition numbers. Then, it was used as an additional loss term to minimize it, which results in better performance in few-shot classification tasks. They demonstrated that their approach converged rapidly, especially for the first few iterations.

**Questions:**

1. All 5 step training was used for all experiments for both MAML and the proposed method. Since MAML might overfit the initial parameters for 5 step performance, so they learned not to update too much in the first few iterations. I am curious how MAML performs, using 1 step training, and 1 step testing accuracy, which is a fairer comparison I think.

2. Missing MAML results on CUB for step 5 - step 100 in Table 3.

3. Missing comparisons to prior arts. Only MAML baseline is not good enough as time of now. For examples, miniImageNet results, the accuracies are way lower than current state-of-the-arts methods. I am not complaining about not achieving SOTA, but it’s better to be upfront with the readers. Especially, you should have shown the performance of methods related to ‘preconditioning’, e,g, [1-3], you can still argue that your method does not require additional parameters.

4. checklist 3-(c), the answer was yes, but I can’t find any error bars w/ different random seeds. I was expecting the ‘shaded area’ in training curves, e.g., standard deviation or confidence interval, at least for the main curves. The standard deviations you reported in the tables are not w/ different random seeds. It’s average over test tasks from my understanding. So, your answers for checklist 3-(c) should be ‘no’.

5. How did you compute eigenvalues? is it differentiable so that you can easily use them as a part of loss function? no explanation about this in the current texts. Also, are there other related works that are using eigenvalues as a part of loss function?

6. Figure 2-(c), you plotted L_k, which is considered as condition numbers. But, was it approximated (by eq 8) and also surrogate condition numbers (by eq (9))? Since you are using L_k as an additional loss function, it is obvious that this number decreases. I am curious if the real condition number is decreasing.

7. In figure 1, (a) and (b), were they a toy example? or from MAML? If MAML, how did you pick only two dimensions in the parameters? not clear given the current texts.

[1] Meta-Curvature, Park et al., NeurIPS 2019
[2] On Modulating the Gradient for Meta-Learning, Simon et al., ECCV 2020
[3] Meta-learning with warped gradient descent, Flennerhag et al., ICLR 2020

**Ethics Review Area:**

["I don’t know"]

**Limitations:**

The authors addressed some limitations in the main text. and I do not see any negative societal impact.

**Strengths And Weaknesses:**

I generally like the idea of achieving rapid convergence without introducing more parameters. The paper is well written and easy to follow. The idea of using least-square reformulation to easily approximate eigenvalues is quite interesting. However, I have some concerns about experimental results below in the question section. I would revise my score depending on the authors' response.

---

> ### Author Response · Authors · 2022-08-02
> **Response to Reviewer FA57 Part1**
>
> We thank you for the constructive feedback and address your concerns point-by-point in the following:
>
> >_All 5 step training was used for all experiments for both MAML and the proposed method. Since MAML might overfit the initial parameters for 5 step performance, so they learned not to update too much in the first few iterations. I am curious how MAML performs, using 1 step training, and 1 step testing accuracy, which is a fairer comparison I think._
>
> Park et al. (2019) provide analyses regarding the 1-step results for MAML and demonstrate that 5-step training and testing notably outperforms the 1-step experiments: 63.92% vs. 59.26% on 5-way 5-shot and 48.85% vs. 46.28% on 5-way 1-shot scenarios for 5-steps vs. 1-step, respectively (Table 3 in their paper, using a Conv4 architecture on the miniImageNet dataset). We thus adopted the comparison to the higher-performing versions of MAML for fairness, i.e. the 5-step approach. Note that both the 5-step and 1-step are outperformed by our method, achieving accuracies of 65.26% and 48.94% on 5-shot and 1-shot settings, respectively.
>
> >_Missing MAML results on CUB for step 5 - step 100 in Table 3._
>
> Due to space limitations, we were unfortunately not able to include all experimental results into main body of the paper, but detailed results for both our conditioned version and the original MAML across all backbones and datasets are provided in the supplementary material in Table A1 (for steps 1-5) and Table A2 (for steps 5-100).
>
> >_Missing comparisons to prior arts. Only MAML baseline is not good enough as time of now. For examples, miniImageNet results, the accuracies are way lower than current state-of-the-arts methods. I am not complaining about not achieving SOTA, but it’s better to be upfront with the readers. Especially, you should have shown the performance of methods related to ‘preconditioning’, e,g, [1-3], you can still argue that your method does not require additional parameters._
>
> As pointed out in the question, achieving state of the art results has not been the objective of this paper, but rather the demonstration of how the condition number of the approximated Hessian can be helpful in learning environments, the empirical findings that the condition number of certain parameter subsets seems to be a good predictor for the overall condition number of the network, its correlation with the few-step adaptation performance, and others. We chose MAML as a basis due to its popularity and simplicity, which provides a well-suited basis for such analyses (without interference of too many added complexities). We do however agree that comparison to other state of the art preconditioning methods (additional parameters and achieved performance) will indeed prove helpful to determine the best method for any individual use case, provide an initial comparison in the table below and propose to include this together with the respective discussion into the supplementary material of our revised paper (supplemented by a reference in the main paper pointing towards these results and the discussion, due to space limitations).
> | Method | Inner Loop Parameter Update|  Add. Params | Test Acc. |
> |:----------|:-------------|:------:|:----:|
> | MAML [A] |  $\boldsymbol{\theta}^{(k)}\_{\tau} = \boldsymbol{\theta}^{(k-1)}\_{\tau} - \alpha \nabla\_{\boldsymbol{\theta}^{(k-1)}}\mathcal{L}\left(\boldsymbol{\theta}^{(k-1)}\_{\tau}\right) $ | -- | $63.1\pm0.9$ |
> | Ours |    $\boldsymbol{\theta}^{(k)}\_{\tau} = \boldsymbol{\theta}^{(k-1)}\_{\tau} - \alpha \nabla\_{\boldsymbol{\theta}^{(k-1)}}\mathcal{L}\left(\boldsymbol{\theta}^{(k-1)}\_{\tau}\right) $ | -- | $65.3\pm0.7$|
> | Meta-SGD [B]  | $\boldsymbol{\theta}^{(k)}\_{\tau} = \boldsymbol{\theta}^{(k-1)}\_{\tau} - \alpha  \mathrm{diag}(\phi) \nabla\_{\boldsymbol{\theta}^{(k-1)}}\mathcal{L}(\boldsymbol{\theta}^{(k-1)}\_{\tau})$ | $\mathrm{diag}(\phi)$ | $64.0±0.9$ |
> | MC [C]|$\boldsymbol{\theta}^{(k)}\_{\tau} = \boldsymbol{\theta}^{(k-1)}\_{\tau} - \alpha M(\boldsymbol{\theta}^{(k-1)}\_{\tau},\psi) \nabla\_{\boldsymbol{\theta}^{(k-1)}}\mathcal{L}(\boldsymbol{\theta}^{(k-1)}\_{\tau})$ |$\psi$ | $68.0\pm0.7$ |
> | ModGrad [D]|$\boldsymbol{\theta}^{(k)}\_{\tau} = \boldsymbol{\theta}^{(k-1)}\_{\tau} - \alpha M\_{\tau}^{(k-1)}(\Psi) \nabla\_{\boldsymbol{\theta}^{(k-1)}}\mathcal{L}(\boldsymbol{\theta}^{(k-1)}\_{\tau})$ | $\Psi$ | $69.2\pm0.7$ |
> | Warp-MAML [E]|$\boldsymbol{\theta}^{(k)}\_{\tau} = \boldsymbol{\theta}^{(k-1)}\_{\tau} - \alpha \nabla\_{\boldsymbol{\theta}^{(k-1)}}\mathcal{L}(\boldsymbol{\theta}^{(k-1)}\_{\tau}, \zeta)$|$\zeta$ | $68.4\pm0.6$|
>
> [A] Finn et al.: Model-agnostic meta-learning for fast adaptation of deep networks. ICML, 2017
>
> [B] Li et al.: Meta-SGD: Learning to Learn Quickly for Few Shot Learning. arXiv:1707.09835, 2017
>
> [C] Park et al.: Meta-Curvature. NeurIPS, 2019
>
> [D] Simon et al.: On Modulating the Gradient for Meta-Learning. ECCV, 2020
>
> [E] Flennerhag et al.: Meta-Learning with Warped Gradient Descent. ICLR, 2020

---

> > ### Author Response · Authors · 2022-08-02
> > **Response to Reviewer FA57 Part2**
> >
> > ... continued from previous
> >
> > > _[...]  your answers for checklist 3-(c) should be ‘no’._
> >
> > While we have run our initial experiments (small architectures) for three different seeds, it turned out not to be feasible for the entire number of experiments given our limited computational resources. We observed in these experiments that training our method proved very stable/repeatable with little variation in the outcomes. To provide some insight, training a Conv6 architecture with three different random initializations on miniImageNet using our method yields top mean validation accuracies of 69.79%, 69.76% and 69.82%. We have further investigated different seeds for test evaluations and did not observe major differences that would not support our conclusions drawn from the single runs (see table below). We are happy to include some training curves with error bars into the supplementary material to demonstrate the stability of our training method, if the reviewers consider this data to be helpful/supportive. Nevertheless, we agree that despite these insights the fitting answer to 3-(c) considering the reported results should
> > be ‘no’.
> > | Conv6 | step 1|  step 2 |  step 3 |  step 4 |  step 5 |
> > |:---|:-----:|------:|:-----:|:-----:|:-----:|
> > |seed1|  63.57±0.73| 67.05±0.73| 68.12±0.72| 68.58±0.73| 68.88±0.73|
> > |seed2| 62.75±0.69 |66.96±0.70| 67.77±0.69| 68.08±0.70| 68.31±0.70|
> > |seed3| 62.38±0.70| 66.33±0.71| 67.49±0.70| 67.74±0.70| 68.05±0.70|
> > |seed4| 62.26±0.68| 66.44±0.70| 67.40±0.70| 67.73±0.71| 67.88±0.71|
> > |seed5| 62.53±0.68 |67.09±0.68 |67.96±0.68| 68.52±0.67| 68.78±0.68|
> > reported| 62.31±0.72| 66.66±0.71| 67.63±0.72| 68.21±0.71| 68.43±0.7|
> >
> > > _How did you compute eigenvalues? is it differentiable so that you can easily use
> > them as a part of loss function?_
> >
> > We compute the eigenvalues in fully differentiable form via eigendecomposition (ED).
> > Our approximation of the Hessian via $JJ^\top$ (eq. (8)) yields a real symmetric matrix for which the eigendecomposition always exists and can be written as $H = U \Lambda U^\top$, with both $U$ and $\Lambda$ real-valued. The work of Ionescu [F] is worth
> > noting, where the authors lay out the background of computing the partial derivatives in matrix form for our case of the eigendecomposition of a real symmetric matrix in their Proposition 2 (equations (12) - (14)).
> > Deep learning libraries like PyTorch include several methods to compute the decomposition in a stable and fully differentiable manner. We use the ‘eigh()’ method from the ‘linalg’ library which provides an efficient way of computation for an entire batch of real symmetric matrices (one computation to retrieve the eigenvalues of all update steps at once) and compared favorably regarding speed to other methods in our timing tests.
> >
> > [F] Ionescu et al.: Matrix backpropagation for deep networks with structured layers, ICCV 2015
> >
> > >_Also, are there other related works that are using eigenvalues as a part of loss function?_
> >
> > Approaches like [G,H] propose losses that are inspired by or based on the specific functionality/meaning eigenvalues have in their respective scenarios (e.g. their importance in projections/multi-view geometry). We are, however, not aware of any other approach using eigenvalues within the actual loss function.
> >
> > [G] Zheng Dang et al.: Eigendecomposition-free training of deep networks with zero eigenvalue-based losses, ECCV 2018
> > [H] Kwang Moo Yi et al.: Learning to find good correspondences, CVPR 2018
> >
> > > _Figure 2-(c), you plotted L_k, which is considered as condition numbers. But, was it approximated (by eq 8) and also surrogate condition numbers (by eq (9))? Since you are using L_k as an additional loss function, it is obvious that this number decreases. I am curious if the real condition number is decreasing._
> >
> > We apologize if this has been unclear. As indicated in the description of the y-axis and the figure caption, Figure 2-(c) shows the progress of the **actual ‘real’ condition number**, i.e. max(ev)/min(ev), of the approximated Hessian of our network – displayed in log10 scale simply
> > for better visualization, not the loss. The Hessian is approximated as introduced in eq (8) via the Jacobian product. The decrease is thus not directly caused due to the enforced loss, but instead rather shows that our loss is indeed able to significantly reduce the actual condition number of the network’s parameters. We provide additional insights about the progress of the condition number computed for the parameters of each inner-loop update step in the supplementary material in Figure 3.
> >
> > >In figure 1, (a) and (b), were they a toy example? or from MAML? If MAML, how did you pick only two dimensions in the parameters?_
> >
> > The experiment presented in the motivational Figure 1 is indeed a toy example that we
> > have chosen to visualize the effect of a higher and lower condition number on optimization problems
> > solved with first-order methods. We have updated the Figure’s caption to clarify this.

---

> > > ### Comment · Reviewer_FA57 · 2022-08-09
> > > **Thanks for your detailed response!**
> > >
> > > Thanks for your response. I do appreciate it. My concerns are partially resolved and I will increase my score from 4 to 5, but I still think this paper is at the borderline for NeurIPS. I strongly encourage the authors to perform the apples-to-apple comparison against other recent methods and present the results in the final version if this gets in.
> > >
> > > 1. Please include the comparisons against other preconditioning-based methods, e.g. MC, ModGrad, Warp-MAML, which all of them achieved better performance than the suggested method. You can also present the number of parameters required for those methods as you did in the bs8R's response.
> > >
> > > 2. Please release the code publicly. Currently, no codes are available, but I believe a few lines of codes are enough to implement the suggested method, which can be easily plugged into many existing open-sourced meta-learning software.

---

### Meta-Review · Area_Chair_oWsc · 2022-08-26

**Recommendation:** Accept
**Confidence:** Certain

**Metareview:**

This paper was quite well received by reviewers, with scores of 5, 6, 6, 8.
Reviewers felt the paper was well written, clear and expressed an interesting core idea.
Experimental results compare against MAML and show clear improvements.

The key idea here was inspired by preconditioning, and the method here aims to increase adaptation speed for gradient-based meta-learning methods without incurring extra parameters. The paper recasts the optimisation to a non-linear least-squares formulation and propose a way to enforce a well-conditioned parameter space for meta-learning through the condition number and local curvature perspectives. Experiments show that the approach outperforms unconstrained optimization significantly and does particularly well during initial adaptation phase of optimization.

The AC recommends acceptance.

**Award:**

No

---

### Decision · Program_Chairs · 2022-09-14

Accept